# Quantum Speedup for Hypergraph Sparsification

**Chenghua Liu** [1 2]  **Minbo Gao** [1 2]  **Zhengfeng Ji** [3]  **Mingsheng Ying** [4]

## Abstract

Graph sparsification serves as a foundation for many algorithms, such as approximation algorithms for graph cuts and Laplacian system solvers. As its natural generalization, hypergraph sparsification has recently gained increasing attention, with broad applications in graph machine learning and other areas. In this work, we propose the *first* quantum algorithm for hypergraph sparsification, addressing an open problem proposed by Apers & de Wolf (2020). For a weighted hypergraph with $n$ vertices, $m$ hyperedges, and rank $r$, our algorithm outputs a near-linear size $\varepsilon$-spectral sparsifier in time $\widetilde{O}(r\sqrt{mn}/\varepsilon)$[1]. This algorithm matches the quantum lower bound for constant $r$ and demonstrates quantum speedup when compared with the state-of-the-art $\widetilde{O}(mr)$-time classical algorithm. As applications, our algorithm implies quantum speedups for computing hypergraph cut sparsifiers, approximating hypergraph mincuts and hypergraph $s$-$t$ mincuts.

## 1. Introduction

Sparsification serves as a foundational algorithmic paradigm wherein a densely constructed entity is transitioned to a sparse counterpart while preserving its inherent characteristics. Such a process invariably enhances various facets of algorithmic efficiency, from reduced execution time to optimized space complexity, and even more streamlined communication. A typical instance of this paradigm is the graph sparsification, where the objective is to reduce the number of edges by adjusting edge weights while (approximately) preserving the spectral properties of the original graph. Over the past two decades, graph sparsification has experienced several breakthroughs (see Spielman & Srivastava (2011); Batson et al. (2012)), ultimately leading to the discovery of algorithms that finds spectral sparsifiers of linear size in nearly-linear time (Lee & Sun, 2018). Graph sparsification algorithms are used for crucial tasks such as Laplacian system solver (Cohen et al., 2014), computing random walk properties (Cohen et al., 2016), and solving maximum flow problem (Chen et al., 2022). Additionally, they have found widespread applications in machine learning domains, including computer vision (Simonovsky & Komodakis, 2017), clustering (Peng et al., 2015; Laenen & Sun, 2020; Agarwal et al., 2022), and streaming machine learning algorithms (Braverman et al., 2021).

Hypergraphs, a generalization of graphs, naturally emerge in various real-world scenarios where interactions go beyond pairwise relationships, such as group dynamics in biochemistry, social networks, and trade networks, as they allow a single edge to connect any number of vertices (the max number of vertices an edge contains is called the rank). Thus, as a natural generalization of the garph sparsification, the task of sparsifying hypergraphs are investigated by many researchers, starting from Soma & Yoshida (2019). Hypergraph sparsification significantly reduces the computational cost of calculating hypergraph energy, a crucial quantity for many machine learning tasks, including clustering (Zhou et al., 2006; Hein et al., 2013; Takai et al., 2020), semi-supervised learning (Hein et al., 2013; Zhang et al., 2020; Yadati et al., 2019; Li et al., 2020), and link prediction (Yadati et al., 2020).

With the rapid development of quantum computing, many graph algorithms and machine learning tasks have benefited from quantum speedups. In the context of the sparsification paradigm, the groundbreaking work of Apers & de Wolf (2020) introduced the *first quantum* speedup for graph sparsification with nearly optimal query complexity, showcasing the potential of quantum computing for sparsification tasks. They further proposed several open questions about the potential for quantum speedups in broader sparsification tasks. Among these, a natural and important question is:

*Is there a hypergraph sparsification algorithm*

---

[1]Key Laboratory of System Software (Chinese Academy of Sciences) and State Key Laboratory of Computer Science, Institute of Software, Chinese Academy of Sciences, China [2]University of Chinese Academy of Sciences, Beijing, China [3]Department of Computer Science and Technology, Tsinghua University, Beijing, China [4]Centre for Quantum Software and Information, University of Technology Sydney, Sydney, Australia. Correspondence to: Zhengfeng Ji <jizhengfeng@tsinghua.edu.cn>.

*Proceedings of the 42$^{nd}$ International Conference on Machine Learning*, Vancouver, Canada. PMLR 267, 2025. Copyright 2025 by the author(s).

[1]We use $\widetilde{O}(f)$ to represent $O\left(f \cdot \mathrm{polylog}(m, n, r, 1/\varepsilon)\right)$ throughout this paper to suppress polylogarithmic factors.

*that enables quantum speedups?*

In this paper, we give an affirmative answer to this question. Specifically, we develop a quantum algorithm for constructing an $\varepsilon$-spectral sparsifier of near-linear size for a weighted hypergraph with $n$ vertices, $m$ hyperedges, and rank $r$. Our algorithm achieves near-linear output size—matching the current best classical algorithm—while operating in sublinear time $\widetilde{O}(r\sqrt{mn}/\varepsilon)^2$, which improves the time complexity $\widetilde{O}(mr)$ of the best known classical algorithm[3]. For the constant rank $r$, this time complexity matches the quantum lower bound of $\widetilde{\Omega}(\sqrt{mn}/\varepsilon)$ established by Apers & de Wolf (2020) and contrasts with the classical lower bound of $\Omega(m)$ (see Remark 4.3). Additionally, for dense hypergraphs, where $m \in \Omega(n^r)$, our algorithm achieves near-quadratic improvement, reducing the time complexity from $\widetilde{O}(rn^r)$ classically to $\widetilde{O}(rn^{(r+1)/2})$ quantumly.

**Hypergraph Sparsification** To extend the graph sparsification task to hypergraphs, we need to generalize the quadratic form of the graph Laplacian. This generalization leads to the concept of hypergraph energy, first introduced by Chan et al. (2018); Yoshida (2019). For a hypergraph $H = (V, E, w)$, the energy of a vector $x \in \mathbb{R}^V$ is defined as a sum over all hyperedges $e \in E$, where each term is the product of the edge weight $w_e$ and the "quadratic form" $Q_e(x)$. Here, $Q_e(x)$ represents the *maximum* squared difference between any two vector components $x_u$ and $x_v$ corresponding to vertices $u$ and $v$ in the hyperedge $e$ (see Definition 2.4 for a formal description). The concept of energy captures the spectral properties of the hypergraph, while its inherent nonlinear structure introduces significant computational challenges (Chan et al., 2018).

The task of hypergraph sparsification aims to reduce the number of hyperedges while maintaining the energy of the original hypergraph, ultimately producing a hypergraph sparsifier (see Definition 2.5). Hypergraph sparsification not only is of great theoretical interest, but also has wide applications in many fields, especially in graph machine learning. Consequently, researchers have been developing increasingly sophisticated and analytically refined algorithms for hypergraph sparsification in recent years. Soma & Yoshida (2019) were the first to demonstrate that an $\varepsilon$-spectral sparsifier of size $\widetilde{O}(n^3/\varepsilon^2)$ could be constructed in time $\widetilde{O}(nmr + n^3/\varepsilon^2)$. Subsequently, Bansal et al. (2019) improved the sparsifier size to $\widetilde{O}(r^3 n/\varepsilon^2)$ with a construction time of $\widetilde{O}(mr^2 + r^3 n/\varepsilon^2)$. Further advancements by

Kapralov et al. (2021) and Kapralov et al. (2022) reduced the sparsifier size to nearly linear, achieving $\widetilde{O}(n/\varepsilon^4)$ in polynomial time. The current state-of-the-art algorithm for hypergraph sparsification, proposed by Jambulapati et al. (2023), produces a sparsifier of size $\widetilde{O}(n/\varepsilon^2)$ in almost linear time $\widetilde{O}(mr)$. Independently and concurrently, Lee (2023) presented a polynomial-time algorithm achieving a sparsifier of the same size. For a detailed comparison of these results, see Table 1.

Prior to the emergence of spectral sparsification, early research focused on a relatively weaker notion called the cut sparsification. In the domain of hypergraphs, extensive research has been conducted on hypergraph cut sparsifiers, yielding significant theoretical and practical advances (Kogan & Krauthgamer, 2015; Chekuri & Xu, 2018). These sparsifiers have proven particularly valuable in efficiently approximating cut minimization problems in hypergraphs, facilitating applications across multiple domains. Notable applications include VLSI circuit partitioning (Alpert & Kahng, 1995; Karypis et al., 1999), optimization of sparse matrix multiplication (Akbudak et al., 2013; Ballard et al., 2016), data clustering algorithms (Li & Milenkovic, 2017; Liu et al., 2021), and ranking data analysis (Li & Milenkovic, 2017).

**Main Results** In this work, we propose the first quantum algorithm for hypergraph sparsification that produces a sparsifier of size $\widetilde{O}(n/\varepsilon^2)$ in $\widetilde{O}(r\sqrt{mn}/\varepsilon)$ time, which breaks the linear barrier of classical algorithms.

**Theorem 1.1** (Informal version of Theorem 4.1). *There exists a quantum algorithm that, given query access to a hypergraph $H = (V, E, w)$ with $|V| = n$, $|E| = m$, $w \in \mathbb{R}^E_{\geq 0}$, rank $r$ and $\varepsilon > 0$, outputs with high probability[4] an $\varepsilon$-spectral sparsifier of $H$ with $\widetilde{O}(n/\varepsilon^2)$ hyperedges, in time $\widetilde{O}(r\sqrt{mn}/\varepsilon)$.*

As a corollary, our algorithm could be used to construct a cut sparsifier for a hypergraph in sublinear time. This enables quantum speedups for approximating hypergraph mincuts and $s$-$t$ mincuts, achieving sublinear time complexity with respect to the number of hyperedges. For further details, we refer readers to Section 5.

**Techiques** Our algorithms are inspired by a sampling-based framework appearing in classical hypergraph sparsification algorithms. The overall idea behind the framework is to compute a proper importance weight for each hyperedge, and sample the hyperedges based on the weights.

Specifically, we adopt the method proposed in Jambulapati et al. (2023), where the weight on each hyperedge is

---

[2]To be more precise, our algorithm runs in $\widetilde{O}(r\sqrt{mn}/\varepsilon + r\sqrt{mnr})$ time, of which the second term is usually smaller in most situations. See Remark 4.2 for a more detailed discussion.

[3]Typically, we assume that $\varepsilon \geq \sqrt{n/m}$, as sparsification is only advantageous when the number of hyperedges in the sparsifier is at most $m$.

[4]Throughout this paper, we say something holds "with high probability" if it holds with probability at least $1 - O(1/n)$.

*Table 1.* Summary of results on hypergraph sparsification

| Reference | Type | Sparsifier size | Time Complexity |
|---|---|---|---|
| Soma & Yoshida (2019) | Classical | $O(n^3 \log n/\varepsilon^2)$ | $\widetilde{O}(mnr + n^3/\varepsilon^2)$ |
| Bansal et al. (2019) | Classical | $O(r^3 n \log n/\varepsilon^2)$ | $\widetilde{O}(mr^2 + r^3n/\varepsilon^2)$ |
| Kapralov et al. (2021) | Classical | $nr(\log n/\varepsilon)^{O(1)}$ | $O(mr^2) + n^{O(1)}$ |
| Kapralov et al. (2022) | Classical | $O(n \log^3 n/\varepsilon^4)$ | $\widetilde{O}(mr + \mathrm{poly}(n))$ |
| Jambulapati et al. (2023); Lee (2023) | Classical | $O(n \log n \log r/\varepsilon^2)$ | $\widetilde{O}(mr)^{\dagger}$ |
| This work | Quantum | $O(n \log n \log r/\varepsilon^2)$ | $\widetilde{O}(r\sqrt{mnr} + r\sqrt{mn}/\varepsilon)$ |

$^{\dagger}$ This $\widetilde{O}(mr)$ complexity corresponds to the algorithm proposed in Jambulapati et al. (2023).

set to be the group leverage score overestimate (which we call hyperedge leverage score overestimate in our paper to avoid ambiguity). Classically, computing these overestimates would require $\widetilde{O}(mr)$ time by an iterative/contractive algorithm which mainly follows the algorithm for computing an approximate John ellipse (Cohen et al., 2019). Then, one can sample $\widetilde{O}(n/\varepsilon^2)$ hyperedges in $\widetilde{O}(n/\varepsilon^2)$ time, and reweight them to get a hypergraph sparsifier.

We discover that, with the assistance of a series of classical and quantum techniques, the computation of hyperedge leverage score overestimate can be accelerated. To be more precise, we realize that the computation of hyperedge leverage score overestimate could be executed in a sequence of sparse underlying graphs (see Definition 3.3 for more details), and these sparse underlying graphs can be efficiently constructed (in $\widetilde{O}(r\sqrt{mnr})$ time) by the quantum graph sparification algorithm proposed by Apers & de Wolf (2020). Thus, we obtain a quantum algorithm (Algorithm 1) that allows efficient *queries* to the hyperedge leverage score overestimate running in sublinear time.

The quantum procedure described above, however, introduces a new challenge for the sampling step: the exact sampling probabilities cannot be directly accessed. To resolve this issue, we utilize the technique of "preparing many copies of a quantum state" (Hamoudi, 2022) to get $\widetilde{O}(n/\varepsilon^2)$ samples in $\widetilde{O}(r\sqrt{mn}/\varepsilon)$ time without explicitly computing the normalization constant for the sampling probability. After sampling hyperedges, we encounter a problem in the reweighting stage due to the unknown normalization constant. We address this issue partially using the quantum sum estimation procedure (Theorem 2.9), while the imprecision introduced during this procedure necessitates a more rigorous analysis. We complete the correctness analysis by employing the novel chaining argument proposed in Lee (2023).

**Related Works** Quantum algorithms have demonstrated significant potential in graph-theoretic and optimization

tasks through the sparsification paradigm, offering both theoretical advances and practical applications.

The seminal work of Apers & de Wolf (2020) established quantum algorithms for graph sparsification, demonstrating speedups in cut approximation, effective resistance computation, spectral clustering, and Laplacian system solving, while also providing fundamental lower bounds for quantum graph sparsification. This work catalyzed several breakthrough results for graph problems. Apers & Lee (2021) developed a quantum algorithm for exact minimum cut computation, achieving quantum speedups when the graph's weight ratio is bounded. Apers et al. (2024) extended these techniques to solve the exact minimum $s$-$t$ cut problem. The versatility of quantum graph sparsification was further demonstrated in Cade et al. (2023), where it enabled accelerated motif clustering computations.

A significant theoretical advancement emerged in Apers & Gribling (2024), which generalized the quantum graph sparsification framework to quantum spectral approximation by combining leverage score sampling with Grover search. This generalization enabled efficient approximation of Hessians and gradients in barrier functions for interior point methods, yielding quantum speedups for linear programming under specific conditions. These spectral approximation techniques have found recent applications in various machine learning problems. Song et al. (2023) and Li et al. (2024) applied the quantum spectral approximation and leverage score sampling algorithms to achieve quantum advantages in linear regression and John ellipsoid approximation, respectively.

## 2. Preliminaries

### 2.1. Notation

For clarity, we use $[n]$ to represent the set $\{1, 2, \ldots, n\}$ and $[n]_0$ to represent the set $\{0, 1, \ldots, n-1\}$. Throughout this paper, we use $G = (V, F, c)$ to denote a graph, where

$V$ is the vertex set, $F$ is the edge set, and $c : F \to \mathbb{R}_{\geq 0}$ represents the edge weights. For an undirected weighted hypergraph, we denote it as $H = (V, E, w)$, where $V$ is the vertex set, $E$ is the hyperedge set, and $w : E \to \mathbb{R}_{\geq 0}$ represents the hyperedge weights. We use $n, m$ to denote the size of $V$ and $E$ respectively. Typically, we denote edges (consisting of two vertices) with $f$ and $g$, while reserving $e$ for hyperedges. Given a hyperedge $e$, we use $\binom{e}{2}$ to denote the corresponding induced edge set $\{f \subseteq e : |f| = 2\}$.

## 2.2. Laplacian and Graph Sparsification

For an undirected weighted graph $G = (V, F, c)$, the weighted degree of vertex $i$ is defined by

$$\deg(i) := \sum_{f \in F : i \in f} c_f,$$

where the sum is taken over all edges $f$ that contain $i$, and $c_f$ represents the weight of edge $f$.

**Definition 2.1** (Laplacian). The Laplacian of a weighted graph $G = (V, F, c)$ is defined as the matrix $L_G \in \mathbb{R}^{V \times V}$ such that

$$(L_G)_{ij} = \begin{cases} \deg(i) & \text{if } i = j, \\ -c_{ij} & \text{if } \{i, j\} \in F, \\ 0 & \text{otherwise.} \end{cases}$$

The Laplacian of graph $G$ is given by $L_G = D_G - A_G$, with $A_G$ the weighted adjacency matrix $(A_G)_{ij} = c_{ij}$ and $D$ the diagonal weighted degree matrix $D_G = \text{diag}(\deg(i) : i \in V)$. $L_G$ is a positive semidefinite matrix whenever weight function $c$ is nonnegative. The quadratic form of $L_G$ can be written as

$$x^\top L_G x = \sum_{\{i,j\} \in F} c_{ij} \cdot (x_i - x_j)^2 \qquad (1)$$

for arbitrary vector $x \in \mathbb{R}^V$. Graph sparsification produces a reweighted graph with fewer edges, known as a graph (spectral) sparsifier. A graph spectral sparsifier of $G$ is a reweighted subgraph that closely approximates the quadratic form of the Laplacian for any vector $x \in \mathbb{R}^V$.

**Definition 2.2** (Graph Spectral Sparsifier). Let $G = (V, F, c)$ be a weighted graph. A re-weighted graph $\widetilde{G} = (V, \widetilde{F}, \widetilde{c})$ is a subgraph of $G$, where $\widetilde{c} : \widetilde{F} \to \mathbb{R}_{\geq 0}$ and $\widetilde{F} = \{f \in F : \widetilde{c}_f > 0\}$. For any $\varepsilon > 0$, $\widetilde{G}$ is an $\varepsilon$-spectral sparsifier of $G$ if for any vector $x \in \mathbb{R}^V$, the following holds:

$$\left| x^\top L_{\widetilde{G}} x - x^\top L_G x \right| \leq \varepsilon \cdot x^\top L_G x.$$

In the groundbreaking work by Spielman & Srivastava (2011), the authors demonstrated that graphs can be efficiently sparsified by sampling edges with weights roughly proportional to their effective resistances. This importance sampling approach is foundational to graph sparsification and has also inspired advancements in hypergraph sparsification. Next, we define the effective resistance.

**Definition 2.3** (Effective Resistance). Given a graph $G = (V, F, c)$, the effective resistance of a pair of $i, j \in V$ is defined as

$$R_{ij} := (\delta_i - \delta_j)^\top L_G^+ (\delta_i - \delta_j) = \|L_G^{+/2}(\delta_i - \delta_j)\|^2,$$

where $L_G^+$ denotes the Moore-Penrose inverse of $L_G$, and $\delta_i$ denotes the vector with all elements equal to 0 except for the $i$-th being 1.

For further details, including key properties of effective resistance used in this work, we refer the readers to Appendix A.

## 2.3. Hypergraph Sparsification

Here, we formally define the fundamental concept in hypergraph sparsification, namely the energy.

**Definition 2.4** (Energy). Let $H = (V, E, w)$ be a weighted hypergraph. For every vector $x \in \mathbb{R}^V$, we define its associated energy in $H$ as

$$Q_H(x) := \sum_{e \in E} w_e \cdot Q_e(x), \qquad (2)$$

where $Q_e(x) := \max_{\{i,j\} \subseteq e} (x_i - x_j)^2$.

In the special case when the rank of $H$ is 2, (meaning $H$ is actually a graph), the energy reduces to the quadratic form of graph Laplacian (see Equation (1)).

Similar to graph sparsification, the goal of hypergraph sparsification is to produce a hypergraph spectral sparsifier with fewer hyperedges. The hypergraph spectral sparsifier of hypergraph $H$ is a reweighted subgraph of $H$ that approximately preserves the energy for any vector $x \in \mathbb{R}^V$.

**Definition 2.5** (Hypergraph Spectral Sparsifier). Let $H = (V, E, w)$ be a weighted hypergraph. A re-weighted hypergraph $\widetilde{H} = (V, \widetilde{E}, \widetilde{w})$ is a subgraph of $H$, where $\widetilde{w} : \widetilde{E} \to \mathbb{R}_{\geq 0}$ and $\widetilde{E} = \{e \in E : \widetilde{w}_e > 0\}$. For any $\varepsilon > 0$, $\widetilde{H}$ is an $\varepsilon$-spectral sparsifier of $G$ if for any vector $x \in \mathbb{R}^V$, the following holds:

$$\left| Q_H(x) - Q_{\widetilde{H}}(x) \right| \leq \varepsilon \cdot Q_H(x).$$

The hypergraph cut sparsifier is a weaker notion of sparsification than the spectral sparsifier. Specifically, for a weighted hypergraph $H(V, E, w)$, we restrict $x \in \mathbb{R}^V$ to be the characteristics vector $1_S \in \{0, 1\}^V$ of a vertex subset $S \subseteq V$. The energy $Q_H(1_S)$, or simply $Q_H(S)$, can be expressed as $Q_H(S) = \sum_{e \in \delta_S} w_e$, where $\delta_S$ denotes the

set of hyperedges crossing the cut $(S, V \setminus S)$. The $\varepsilon$-*cut sparsifier* $\widetilde{H}$ of the hypergraph $H$ is a subgraph that satisfies the following:

$$\left| Q_H(S) - Q_{\widetilde{H}}(S) \right| \leq \varepsilon \cdot Q_H(S), \quad \forall S \subseteq V. \quad (3)$$

### 2.4. Quantum Computing and Speedup

In quantum mechanics, a $d$-dimensional quantum state $|v\rangle = (v_0, \ldots, v_{d-1})^\top$ is a unit vector in a complex Hilbert space $\mathbb{C}^d$, namely, $\sum_{i \in [d]_0} |v_i|^2 = 1$. We define the computational basis of the space $\mathbb{C}^d$ by $\{|i\rangle\}_{i \in [d]_0}$, where $|i\rangle = (0, \ldots, 0, 1, 0, \ldots, 0)^\top$ with the $i$-th entry (0-indexed) being 1 and others being 0. The inner product of quantum states $|u\rangle, |v\rangle \in \mathbb{C}^d$ is defined by $\langle u|v\rangle = \sum_{i \in [d]_0} u_i^* v_i$, where $z^*$ denotes the conjugate of $z \in \mathbb{C}$. The tensor product of quantum states $|u\rangle \in \mathbb{C}^{d_1}$ and $|v\rangle \in \mathbb{C}^{d_2}$ is their Kronecker product, $|u\rangle \otimes |v\rangle = (u_0 v_0, u_0 v_1, \ldots, u_{d_1-1} v_{d_2-1})^\top$, which can be abbreviated as $|u\rangle |v\rangle$.

A quantum bit, or qubit, is a quantum state $|\psi\rangle$ in $\mathbb{C}^2$, expressible as $|\psi\rangle = \alpha |0\rangle + \beta |1\rangle$, where $\alpha, \beta \in \mathbb{C}$ and $|\alpha|^2 + |\beta|^2 = 1$. Furthermore, an $n$-qubit state is in the tensor product space of $n$ Hilbert spaces $\mathbb{C}^2$, denoted as $(\mathbb{C}^2)^{\otimes n} = \mathbb{C}^{2^n}$, with the computational basis $\{|i\rangle\}_{i \in [2^n]_0}$. To extract classical information from an $n$-qubit state $|\psi\rangle$, we measure it in the computational basis, yielding outcome $i$ with probability $p(i) = |\langle \psi|i\rangle|^2$ for $i \in [2^n]_0$. The operations in quantum computing are described by unitary matrices $U$, satisfying $UU^\dagger = U^\dagger U = I$, where $U^\dagger$ is the Hermitian conjugate of $U$, and $I$ is the identity matrix.

We consider the following edge-vertex incidence oracle $\mathcal{O}_G$ for the graph $G = (V, F, c)$ with $n$ vertices and $m$ edges. This oracle consists of two unitaries, $\mathcal{O}_G^{\text{vtx}}$ and $\mathcal{O}_G^{\text{wt}}$, which are defined as follows for any edge $f = \{i, j\} \in F$:

$$\mathcal{O}_G^{\text{vtx}} : |f\rangle |0\rangle \mapsto |f\rangle |i\rangle |j\rangle,$$
$$\mathcal{O}_G^{\text{wt}} : |f\rangle |0\rangle \mapsto |f\rangle |c_f\rangle,$$

where $|f\rangle \in \mathbb{C}^m, |i\rangle, |j\rangle \in \mathbb{C}^n$, and $c_f$ is represented as a floating-point number with $|c_f\rangle \in \mathbb{C}^{d_{\text{acc}}}$. Taking $d_{\text{acc}} = \widetilde{O}(1)$ allows for achieving arbitrary desired floating-point accuracy. Similarly, we assume access to hyperedge oracle $\mathcal{O}_H$ for the hypergraph $H = (V, E, w)$, which consists of three unitaries $\mathcal{O}_H^{\text{size}}, \mathcal{O}_H^{\text{vtx}}$ and $\mathcal{O}_H^{\text{wt}}$. These unitaries allow for the following queries for any hyperedge $e \in E$:

$$\mathcal{O}_H^{\text{size}} : |e\rangle |0\rangle \mapsto |e\rangle ||e|\rangle,$$
$$\mathcal{O}_H^{\text{vtx}} : |e\rangle |0\rangle^{\otimes r} \mapsto |e\rangle \left( \bigotimes_{i \in e} |i\rangle \right) |0\rangle^{\otimes (r-|e|)},$$
$$\mathcal{O}_H^{\text{wt}} : |e\rangle |0\rangle \mapsto |e\rangle |w_e\rangle.$$

In many quantum algorithms, information can be stored and retrieved in quantum-read classical-write random access memory (QRAM) (Giovannetti et al. (2008)), which is employed in numerous time-efficient quantum algorithms. QRAM enables the storage or modification of an array $c_1, \ldots, c_n$ of classical data while allowing quantum query access via a unitary $U_{\text{QRAM}} : |i\rangle |0\rangle \mapsto |i\rangle |c_i\rangle$. Although QRAM is a natural quantization of the classical RAM model and is widely utilized, it is important to acknowledge that, given the current advancements of quantum computers, the feasibility of implementing practical QRAM remains somewhat speculative.

A quantum (query) algorithm $\mathcal{A}$ is a quantum circuit consisting of a sequence of unitaries $U_1, \ldots, U_T$, where each $U_t$ could be a quantum gate, a quantum oracle, or a QRAM operation. The time complexity of $\mathcal{A}$ is determined by the number $T$ of quantum gates, oracles and QRAM operations it contains. The algorithm $\mathcal{A}$ operates on $n$ qubits, starting with the initial state $|0\rangle^{\otimes n}$. The unitary operators $U_1, \ldots, U_T$ are then applied sequentially to the quantum state, resulting in the final quantum state $|\psi\rangle = U_T \ldots U_1 |0\rangle^{\otimes n}$. Finally, a measurement is performed on $|\psi\rangle$ in the computational basis $|i\rangle$ for $i \in [2^n]_0$, yielding a classical outcome $i$ with probability $|\langle i|\psi\rangle|^2$.

In our paper, we incorporate quantum algorithms from previous research as basic components of our algorithm. One such algorithm is quantum graph sparsification, initially proposed by Apers & de Wolf (2020) using adjacency-list input queries, and later revisited through alternative techniques in Apers & Gribling (2024) using edge-vertex incidence queries. The query access described below refers to the latter approach.

**Theorem 2.6** (Quantum Graph Sparsification, Theorem 1 in Apers & de Wolf (2020)). *There exists a quantum algorithm* GraphSparsify$(\mathcal{O}_G, \varepsilon)$ *that, given query access $\mathcal{O}_G$ to a weighted graph $G = (V, F, c)$ with $|V| = n, |F| = m, c \in \mathbb{R}_{\geq 0}^F$ and $\varepsilon \geq \sqrt{n/m}$, outputs with high probability the explicit description of an $\varepsilon$-spectral sparsifier of $G$ with $\widetilde{O}(n/\varepsilon^2)$ edges, using $\widetilde{O}(\sqrt{mn}/\varepsilon)$ queries to $\mathcal{O}_G$ and in time $\widetilde{O}(\sqrt{mn}/\varepsilon)$.*

The next quantum algorithm proposed by Hamoudi (2022) provides an efficient approach to prepare many copies of a quantum state.

**Theorem 2.7** (Preparing Many Copies of a Quantum State, Theorem 1 in Hamoudi (2022)). *There exists a quantum algorithm that, given oracle access $\mathcal{O}_w$ to a vector $w \in \mathbb{R}_{\geq 0}^n$ (0-indexed) ($\mathcal{O}_w : |i\rangle |0\rangle \mapsto |i\rangle |w_i\rangle, \forall i \in [n]$), and $k \in [n]$, with high probability, outputs $k$ copies of the state $|w\rangle$, where*

$$|w\rangle = \frac{1}{\sqrt{W}} \sum_{i \in [n]_0} \sqrt{w_i} |i\rangle$$

*with $W = \sum_{i \in [n]_0} w_i$. The algorithm uses $\widetilde{O}(\sqrt{nk})$ queries to $\mathcal{O}_w$, and runs in $\widetilde{O}(\sqrt{nk})$ time.*

By performing measurements on each of the generated quantum state copies, a sample sequence is produced, where each element $i$ is selected with a probability proportional to $w_i$. This leads to the following corollary.

**Corollary 2.8.** *There exists a quantum algorithm* MultiSample$(\mathcal{O}_w, k)$ *that, given query access $\mathcal{O}_w$ to a vector $w \in \mathbb{R}^n_{\geq 0}$ and integer $k \in [n]$, outputs with high probability a sample sequence $\sigma \in [n]^k$ such that each element $i$ is sampled with probability proportional to $w_i$. The algorithm uses $\widetilde{O}(\sqrt{nk})$ queries to $\mathcal{O}_w$, and runs in $\widetilde{O}(\sqrt{nk})$ time.*

In addition to the quantum algorithms mentioned above, we also require quantum sum estimation, which provides a quadratic speedup over classical approaches.

**Theorem 2.9** (Quantum Sum Estimation, Lemma 3.1 in Li et al. (2019)). *There exists a quantum algorithm* SumEstimate$(\mathcal{O}_w, \varepsilon)$ *that, given query access $\mathcal{O}_w$ to a vector $w \in \mathbb{R}^n_{\geq 0}$ and $\varepsilon > 0$, outputs with high probability an estimate $\widetilde{s}$ for $s = \sum_{i \in [n]} w_i$ satisfying $|\widetilde{s} - s| \leq \varepsilon s$, using $\widetilde{O}(\sqrt{n}/\varepsilon)$ queries to $\mathcal{O}_w$ and in $\widetilde{O}(\sqrt{n}/\varepsilon)$ time.*

# 3. Quantum Algorithm for Leverage Score Overestimates

In this section, we will introduce the notion of hyperedge leverage scores, which is a generalization of leverage scores of edges in a graph. We then define the concept of leverage score overestimates, which are one-side bounded estimates of hyperedge leverage scores. Finally, we propose a quantum algorithm that computes the overestimates given query access to a hypergraph.

To define hyperedge leverage scores, we first introduce the concept of underlying graphs.

**Definition 3.1** (Underlying Graph). Given an undirected weighted hypergraph $H = (V, E, w)$, an underlying graph of $H$ is defined as a multigraph $G = (V, F, c)$ with edge set $F = \left\{(e, f) : f \in \binom{e}{2}, e \in E\right\}$ and weights $c \in \mathbb{R}^F_{\geq 0}$, satisfying

$$w_e = \sum_{f \in \binom{e}{2}} c_{e,f}, \qquad \forall e \in E. \tag{4}$$

Note that if the hypergraph contains $m$ hyperedges with rank $r$, its underlying graph can have up to $mr(r-1)/2$ edges. The multiple edges in the underlying graph are labeled according to the hyperedges they originate from.

With this concept, we define the hyperedge leverage score as follows: Given a hypergraph $H$ and one of its corresponding underlying graphs $G$, the *leverage score of a hyperedge* $e \in E$, is defined as

$$\ell_e := w_e R_e, \tag{5}$$

where $R_e = \max\{R_f : \forall f \in \binom{e}{2}\}$, and $R_f$ represents the effective resistance of the edge $f$ in the underlying graph $G$.

We remark that, the choice of the underlying graph $G$ will greatly influence the hyperedge leverage scores. For our sparsification purpose, we want to bound the total sum of hyperedge leverage scores by $O(n)$, which determines the the size of the resulting hypergraph sparsifiers. Therefore, we define the following notion of hyperedge leverage score overestimates, which are entry-wise upper bounds for leverage scores with a specific underlying graph, such that the total sum is bounded by a parameter $\nu = O(n)$.

**Definition 3.2** (Hyperedge Leverage Score Overestimate, adapted from Jambulapati et al. (2023, Definition 1.3)). Given a hypergraph $H = (V, E, w)$, we say $z \in \mathbb{R}^E_{\geq 0}$ is a $\nu$-(bounded hyperedge leverage score) overestimate for $H$ if $\|z\|_1 \leq \nu$ and there exists a corresponding underlying graph $G = (V, F, c)$, satisfying the constraints Equation (4), such that for all $e \in E$, $z_e \geq \ell_e$.

To obtain an overestimate, we need to handle the weights of the underlying graph, which contains $O(mr^2)$ edges. Nevertheless, it is sufficient to manage only $O(mr)$ edges by replacing each hyperedge with a sparse subgraph (e.g., a star graph with up to $r - 1$ edges) rather than a complete clique (Definition 3.1). We refer to the resulting graph as a sparse underlying graph.

**Definition 3.3** (Sparse Underlying Graph). Given an undirected weighted hypergraph $H = (V, E, w)$, a sparse underlying graph of $H$ is defined as $G = (V, F, c)$ with edge set $F = \{(e, f) : f \in S_e, e \in E\}$ and weights $c \in \mathbb{R}^F_{\geq 0}$, satisfying

$$w_e = \sum_{f \in S_e} c_{e,f}, \qquad \forall e \in E. \tag{6}$$

For each $e \in E$, we fix an arbitrary vertex $a_e \in e$ and define $S_e = \{f \in \binom{e}{2} : a_e \in f\}$.

Now we present our quantum algorithm of hyperedge leverage score overestimates, which is a key step for our main quantum algorithm for hypergraph sparsification. Our algorithm is inspired by the approximating John ellipsoid algorithm proposed in Cohen et al. (2019) and the group leverage score overestimate algorithm in Jambulapati et al. (2023). The input of our algorithm includes a quantum oracle to the hypergraph, the number of iterations $T$, the graph sparsification parameter $\alpha_1$, and the effective resistance approximation factor $\alpha_2$. The output of our quantum algorithm is a data structure, which could provide a query access to the hyperedge leverage score overestimates (see Proposition B.6 for a formal description of the output).

Recall that the choice of the underlying graph $G$ determines the hyperedge leverage scores, as well as their overestimates. Our algorithm iteratively adjusts the edge weights

of the underlying graph over roughly $\log r$ rounds to construct a suitable $G$. In each iteration, we use quantum graph sparsification to reduce the graph's size and reassign hyperedge weights to the edges of the underlying graph according to edge leverage scores. To maintain overall efficiency, this process is implemented entirely quantumly through a series of quantum subroutines. The process begins with WeightInitialize, which is employed during the first iteration to establish quantum query access to the weights of the underlying graph via queries to the original hypergraph (Proposition B.1). In each iteration, the system utilizes UGraphStore to provide quantum query access to the stored weights of the sparsifier obtained from GraphSparsify (Proposition B.4). Subsequently, EffectiveResistance enables efficient quantum queries to the approximate effective resistance of a graph (Proposition B.3). For the next iteration, WeightCompute implements quantum query access to the updated weights of the sparse underlying graph (Proposition B.5). The complete algorithm is presented in Algorithm 1.

---

**Algorithm 1** Quantum Hyperedge Leverage Score Overestimates QHLSO($\mathcal{O}_H, T, \alpha_1, \alpha_2$)

---

**Require:** Quantum Oracle $\mathcal{O}_H$ to a hypergraph $H = (V, E, w)$ with $|V| = n, |E| = m$, rank $r$; the number of episodes $T \in \mathbb{N}$; positive real numbers $\alpha_1, \alpha_2 \in \mathbb{R}$.
**Ensure:** An instance $\mathcal{Z}$ of QOverestimate which stores the vector $z$ being an $O(n)$-overestimate for $H$.
1: Let $U_{G(1)} = $ WeightInitialize($\mathcal{O}_H$).
2: **for** $t = 1$ to $T$ **do**
3:     $\widetilde{G}^{(t)} = (V, \widetilde{F}^{(t)}, \widetilde{c}^{(t)}) \leftarrow$ GraphSparsify($U_{G(t)}, \alpha_1$).
4:     $\mathcal{G}^{(t)} \leftarrow$ UGraphStore($\widetilde{G}^{(t)}$).
5:     $\mathcal{R}^{(t)} \leftarrow$ EffectiveResistance($\widetilde{G}^{(t)}, \alpha_2$).
6:     $U_{G(t+1)} = $ WeightCompute($\mathcal{O}_H, \mathcal{R}^{(t)}, \mathcal{G}^{(t)}$).
7: **end for**
8: $C_1 \leftarrow 2(1 + \frac{\alpha_1 + \alpha_2}{1 - \alpha_1}) \cdot \exp(\log r/T)$.
9: $\mathcal{Z} \leftarrow$ QOverestimate($\{\mathcal{G}^{(t)} : t \in [T]\}, \{\mathcal{R}^{(t)} : t \in [T]\}, \mathcal{O}_H, C_1, T$).

---

The algorithm's complexity is described in the following theorem.

**Theorem 3.4** (Quantum Hyperedge Leverage Score Overestimates). *There exists a quantum algorithm QHLSO($\mathcal{O}_H, T, \alpha_1, \alpha_2$) that, given integer $T = O(\log r)$, positive real numbers $\alpha_1, \alpha_2 < 1$ and query access $\mathcal{O}_H$ to a hypergraph $H = (V, E, w)$ with $|V| = m, |V| = n, w \in \mathbb{R}_{\geq 0}^E$, and rank $r$, the algorithm runs in time $\widetilde{O}(r\sqrt{mnr})$. Then, with high probability, it provides query access to a $\nu$-overestimate $z$ with $\nu = O(n)$, where each query to $z$ requires $\widetilde{O}(r)$ time.*

Due to space constraints, the proof of Theorem 3.4 is deferred to Appendix B.

# 4. Quantum Hypergraph Sparsification

Assuming query access to a $\nu$-overestimate, we aim to implement the sampling scheme in a quantum framework. By leveraging Corollary 2.8, we can sample a sequence $\sigma = (\sigma_i : \sigma_i \in E)$ where each element $e$ is sampled with probability proportional to $z_e$. By combining the information of each sampled $e$ with the normalization factor obtained via SumEstimate, we assign appropriate weights to the sampled edges to construct the final sparsifier. The complete algorithm is outlined in Algorithm 2.

---

**Algorithm 2** Quantum Hypergraph Sparsification QHypergraphSparse($\mathcal{O}_H, \varepsilon$)

---

**Require:** Quantum Oracle $\mathcal{O}_H$ to a hypergraph $H = (V, E, w)$ with $|V| = n, |E| = m$, rank $r$; accuracy $\varepsilon > 0$.
**Ensure:** An $\varepsilon$-spectral sparsifier of $H$, denoted by $\widetilde{H} = (V, \widetilde{E}, \widetilde{w}), |\widetilde{E}| = O(n \log n \log r/\varepsilon^2)$.
1: $\widetilde{E} = \emptyset, \widetilde{w} = 0, M \leftarrow \Theta\left(n \log n \log r/\varepsilon^2\right)$.
2: $\mathcal{Z} \leftarrow$ QHLSO($\mathcal{O}_H, \log(r-1), 0.1, 0.1$).
3: $\sigma \leftarrow$ MultiSample($\mathcal{Z}$.Query, $M$).
4: $s \leftarrow$ SumEstimate($\mathcal{Z}$.Query, 0.1).
5: **for** $i = 1$ to $M$ **do**
6:     $w_{\sigma_i} \leftarrow$ measurement outcome of the second register of $\mathcal{O}_H^{\text{wt}} |\sigma_i\rangle |0\rangle$.
7:     $z_{\sigma_i} \leftarrow$ measurement outcome of the second register of $\mathcal{Z}$.Query $|\sigma_i\rangle |0\rangle$.
8:     $\widetilde{E} \leftarrow \widetilde{E} \cup \{\sigma_i\}, \widetilde{w}_{\sigma_i} \leftarrow \widetilde{w}_{\sigma_i} + w_{\sigma_i} \cdot s/(M z_{\sigma_i})$.
9: **end for**

---

**Theorem 4.1** (Quantum Hypergraph Sparsification). *There exists a quantum algorithm that, given query access to a hypergraph $H = (V, E, w)$ with $|E| = m, |V| = n, w \in \mathbb{R}_{\geq 0}^E$, rank $r$, and $\varepsilon > 0$, outputs with high probability the explicit description of an $\varepsilon$-spectral sparsifier of $H$ with $O(n \log n \log r/\varepsilon^2)$ hyperedges, in time $\widetilde{O}(r\sqrt{mnr} + r\sqrt{mn}/\varepsilon)$.*

The proof of correctness for the algorithm follows closely the chaining argument in Lee (2023) and Jambulapati et al. (2023). Due to space constraints, we provide the detailed proof in Appendix C.

*Remark* 4.2. We assume $\varepsilon \geq \sqrt{n/m}$, as sparsification is beneficial only when the sparsifier contains less $m$ hyperedges. It is also generally the case that $m \geq nr$, as the objects being sparsified are dense. It's worth noting that the time complexity $\widetilde{O}\left(r\sqrt{mnr} + r\sqrt{mn}/\varepsilon\right)$ simplifies to $\widetilde{O}\left(r\sqrt{mn}/\varepsilon\right)$ whenever $\varepsilon \geq \sqrt{n/m}$ and $m \geq nr$.

*Remark* 4.3. For the hypergraph with constant rank $r$, the above complexity contrasts with the classical lower bound of $\Omega(m)$. This lower bound arises from the fact that, in the case of graphs ($r = 2$), there exists an $\Omega(m)$ classical query lower bound for determining whether a graph is con-

nected, which establishes the same lower bound for both cut sparsifiers and spectral sparsifiers.

## 5. Applications

As a direct corollary of Theorem 4.1, we can compute a cut sparsifier for a hypergraph in sublinear time.

**Corollary 5.1** (Quantum Hypergraph Cut Sparsification). *There exists a quantum algorithm that, given query access to a hypergraph $H = (V, E, w)$ with $|E| = m, |V| = n, w \in \mathbb{R}_{\geq 0}^E$, rank $r$, and $\varepsilon > 0$, outputs with high probability the explicit description of an $\varepsilon$-cut sparsifier of $H$ with $O(n \log n \log r / \varepsilon^2)$ hyperedges in time $\widetilde{O}(r\sqrt{mnr} + r\sqrt{mn}/\varepsilon)$.*

Similar to the case for graphs, quantum hypergraph cut sparsification facilitates faster approximation algorithms for cut problems. Below, we highlight two such applications.

**Mincut**  Given a hypergraph $H = (V, E, w)$, the hypergraph mincut problem asks for a vertex set $S : \emptyset \subsetneq S \subsetneq V$ that minimizes the energy $Q_H(S)$. To the best of our knowledge, the fastest algorithm for computing the mincut in a hypergraph without error runs in $\widetilde{O}(mnr)$ time (Klimmek & Wagner, 1996; Mak & Wong, 2000). By applying Corollary 5.1, we first sparsify the hypergraph and then apply the mincut algorithm to the cut sparsifier. This gives a quantum algorithm that, with high probability, outputs a $(1 + \varepsilon)$-approximate mincut in time $\widetilde{O}(r\sqrt{mn}/\varepsilon + rn^2)$, which is sublinear in the number of hyperedges.

**Corollary 5.2** (Quantum Hypergraph Mincut Solver). *There exits a quantum algorithm that, given query access to a hypergraph $H = (V, E, w)$ with $|E| = m, |V| = n, w \in \mathbb{R}_{\geq 0}^E$, rank $r$, and $\varepsilon > 0$, outputs with high probability the $(1 + \varepsilon)$-approximate mincut of $H$ in time $\widetilde{O}(r\sqrt{mnr} + r\sqrt{mn}/\varepsilon + rn^2)$.*

**$s$-$t$ mincut**  Given a hypergraph $H = (V, E, w)$ and two vertices $s, t \in V$, the $s$-$t$ mincut problem seeks a vertex set $S \subseteq V$ with $|S \cap \{s, t\}| = 1$(i.e., either $s \in S$ or $t \in S$) that minimizes the energy $Q_H(S)$. The standard approach for computing an $s$-$t$ mincut in a hypergraph is computing the $s$-$t$ maximum flow in an associated digraph with $O(n + m)$ vertices and $O(mr)$ edges (Lawler, 1973). And an $s$-$t$ maximum flow in such graph can be found in $\widetilde{O}(mr\sqrt{m + n})$ time (Lee & Sidford, 2014). By combining Corollary 5.1 with the aforementioned approach, we can compute a $(1 + \varepsilon)$-approximate $s$-$t$ mincut in time $\widetilde{O}(r\sqrt{mn}/\varepsilon)$ whenever $m = \Omega(n^2)$, which is sublinear in number of hyperedges.

**Corollary 5.3** (Quantum Hypergraph $s$-$t$ Mincut Solver). *There exits a quantum algorithm that, given query access to a hypergraph $H = (V, E, w)$ with $|E| = m, |V| = n, w \in \mathbb{R}_{\geq 0}^E$, rank $r$, two vertices $s, t \in V$, and $\varepsilon > 0$, outputs with high probability the $(1 + \varepsilon)$-approximate $s$-$t$ mincut of $H$ in time $\widetilde{O}(r\sqrt{mnr} + r\sqrt{mn}/\varepsilon + rn^{3/2})$.*

## 6. Conclusion and Future works

In this paper, we present a quantum algorithm for hypergraph sparsification with time complexity $\widetilde{O}(r\sqrt{mn}/\varepsilon + r\sqrt{mnr})$, where $m, n, r, \varepsilon$ are the number of hyperedges, the number of vertices, rank of hypergraph and precision of sparsifier, respectively.

Our paper naturally raises several open questions for future work. For instance:

- Quantum graph sparsification has directly led to the development of numerous quantum algorithms, including max-cut for graphs (Apers & de Wolf, 2020), graph minimum cut finding (Apers & Lee, 2021), graph minimum $s$-$t$ cut finding (Apers et al., 2024), motif clustering (Cade et al., 2023). A natural question arises: for more related problems in hypergraphs, such as hypergraph-$k$-cut, minmax-hypergraph-$k$-partition, hypergraph spectral diffusion (Ameranis et al., 2023), can we design faster quantum algorithms compared to classical ones?

- We conjecture that the runtime of our quantum algorithm is tight up to polylogarithmic factors when $\varepsilon \leq 1/\sqrt{r}$. Can we establish a quantum lower bound of $\Omega(r\sqrt{mn}/\varepsilon)$ for hypergraph sparsification? Alternatively, can we improve the time complexity for quantum hypergraph sparsification, or further enhance the runtime for hypergraph cut sparsification? Furthermore, state-of-the-art hypergraph cut sparsification achieves a size of $O(n \log n/\varepsilon^2)$ (without the $\log r$ factor) in $\widetilde{O}(mn + n^{10}/\varepsilon^7)$ time (Chen et al., 2020)—can we design a faster quantum algorithm that matches this size?

- Hypergraph sparsification has been extended to various frameworks, including online hypergraph sparsification (Soma et al., 2024; Khanna et al., 2025), directed hypergraph sparsification (Oko et al., 2023), submodular hypergraph sparsification (Kenneth & Krauthgamer, 2024), generalized linear models sparsification (Jambulapati et al., 2024), and quotient sparsification for submodular functions (Quanrud, 2024). A natural question is can we develop specialized quantum algorithms for these settings?

## Acknowledgements

The work was supported by National Key Research and Development Program of China (Grant No.

2023YFA1009403), National Natural Science Foundation of China (Grant No. 12347104), Beijing Natural Science Foundation (Grant No. Z220002), and Tsinghua University.

## Impact Statement

This paper presents work whose goal is to advance the field of Machine Learning. There are many potential societal consequences of our work, none of which we feel must be specifically highlighted here.

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

## A. Useful properties of effective resistance

It's the well-known fact in graph theory that the effective resistance defines a metric on the vertices of a graph. Below, we outline several key properties of effective resistance that will be utilized in this work.

**Lemma A.1** (Foster). *For a weighted graph $G = (V, F, c)$, let $R_{ij}$ represent the effective resistance between vertices $i$ and $j$. Then, it holds that $\sum_{\{i,j\} \in F} c_{ij} R_{ij} \leq n$.*

*Proof.* For a edge $\{i, j\}$, recall that $R_{ij} = (\delta_i - \delta_j)^\top L_G^+ (\delta_i - \delta_j)$, then

$$\sum_{\{i,j\} \in F} c_{ij} R_{ij} = \sum_{\{i,j\} \in F} \mathrm{Tr}\left(c_{ij}(\delta_i - \delta_j)(\delta_i - \delta_j)^\top L_G^+\right) = \mathrm{Tr}\left(L_G L_G^+\right) \leq n - 1$$

since rank of $L_G$ is at most $n - 1$. $\qquad\square$

**Lemma A.2** (Convexity, Lemma 3.4 in Cohen et al. (2019)). *For a weighted graph $G = (V, F, c)$, the function $\log R_f(c)$ is convex with respect to $c$.*

## B. Proof of Theorem 3.4

For a hyperedge $e$, let $S_e$ represent the set $\{f \in \binom{e}{2} : a \in f\}$, where $a$ is a fixed vertex in $e$.

**Proposition B.1** (Weight Initialization for Overestimates). *Suppose $H = (V, E, w)$ is a hypergraph with vertex set $V$ of size $n$, edge set $E$ of size $m$, weight function $w : E \to \mathbb{R}_{\geq 0}$, and $\mathcal{O}_H$ is a quantum oracle to $H$. Then, there exists a quantum algorithm $\mathsf{WeightInitialize}(\mathcal{O}_H)$, that satisfies*

$$\mathsf{WeightInitialize}(\mathcal{O}_H) |e\rangle |f\rangle |0\rangle = |e\rangle |f\rangle |c_{e,f}^{(1)}\rangle$$

*with $c_{e,f}^{(1)} = w_e/(|e| - 1)$, for $\forall e \in E$ and $\forall f = \{i, j\} \in S_e$, and performs no action if $f = \{i, j\} \notin S_e$, using $O(1)$ queries to $\mathcal{O}_H$, in $\widetilde{O}(1)$ time. Here, we represent $|f\rangle$ as $|i\rangle |j\rangle$ for $f = \{i, j\}$.*

*Proof.* Consider the following procedure: for any $e \in E$ and $f \in S_e$, we have

$$
\begin{aligned}
|e\rangle |f\rangle |0\rangle |0\rangle |0\rangle &\xmapsto{\mathcal{O}_H^{\mathrm{wt}}, \mathcal{O}_H^{\mathrm{size}}} |e\rangle |f\rangle |w_e\rangle ||e|\rangle |0\rangle \\
&\xmapsto{U_{\mathrm{fwd}}} |e\rangle |f\rangle |w_e\rangle ||e| - 1\rangle |0\rangle \\
&\xmapsto{U_{\mathrm{div}}} |e\rangle |f\rangle |w_e\rangle ||e| - 1\rangle |c_{e,f}^{(1)}\rangle \\
&\xmapsto{U_{\mathrm{fwd}}^\dagger} |e\rangle |f\rangle |w_e\rangle ||e|\rangle |c_{e,f}^{(1)}\rangle \\
&\xmapsto{\mathcal{O}_H^{\mathrm{size}\dagger}, \mathcal{O}_H^{\mathrm{wt}\dagger}} |e\rangle |f\rangle |0\rangle |0\rangle |c_{e,f}^{(1)}\rangle,
\end{aligned}
$$

where $U_{\mathrm{fwd}}$ satisfies $U_{\mathrm{fwd}} |i\rangle = |i - 1\rangle$, $U_{\mathrm{div}}$ satisfies $U_{\mathrm{div}} |i\rangle |j\rangle |0\rangle = |i\rangle |j\rangle |i/j\rangle$. This procedure uses 4 queries to $\mathcal{O}_H$ and $\widetilde{O}(1)$ additional arithmetic operations. Therefore, let $\mathsf{WeightInitialize}(\mathcal{O}_H) = \mathcal{O}_H^{\mathrm{wt}\dagger} \mathcal{O}_H^{\mathrm{size}\dagger} U_{\mathrm{fwd}}^\dagger U_{\mathrm{div}} U_{\mathrm{fwd}} \mathcal{O}_H^{\mathrm{size}} \mathcal{O}_H^{\mathrm{wt}}$, we know it satisfies the requirement stated in the proposition. $\qquad\square$

Recall the following classical algorithm for efficiently computing the effective resistances of a graph proposed by Spielman & Srivastava (2011).

**Theorem B.2** (Effective Resistance Oracle, Theorem 2 in Spielman & Srivastava (2011)). *There exists a (classical) algorithm $\mathsf{ClassicalEffectiveResistance}(G, \varepsilon)$ such that for any $\varepsilon > 0$ and graph $G = (V, F, c)$ with vertex set $V$ of size $n$, edge set $F$ of size $m$, and weights $c \in \mathbb{R}_{\geq 0}^F$, with high probability, returns a matrix $Z_G$ of size $p \times n$ with $p = \lceil 24 \log n / \varepsilon^2 \rceil$ satisfying*

$$(1 - \varepsilon) R_{ab} \leq \|Z_G(\delta_a - \delta_b)\|^2 \leq (1 + \varepsilon) R_{ab}$$

*for every pair of $a, b \in V$, where $R_{ab}$ is the effective resistance between $a$ and $b$, in $\widetilde{O}(m/\varepsilon^2)$ time.*

We require a quantum variant of this effective resistance computation, as outlined below.

**Proposition B.3** (Quantum Effective Resistance Oracle, Apers & de Wolf (2020, Claim 7.9)). *Let $G = (V, F, c)$ be a graph with a vertex set $V$ of size $n$, an edge set $F$ of size $m$, and weights $c :\in \mathbb{R}_{\geq 0}^F$. For $\varepsilon > 0$, there is a quantum data structure* EffectiveResistance, *that supports the following operations:*

- *Initialization:* EffectiveResistance$(G, \varepsilon)$, *outputs an instance $\mathcal{R}$, in $\widetilde{O}(m/\varepsilon^2)$ time.*

- *Query:* $\mathcal{R}$.Query, *outputs a unitary satisfying*

$$\mathcal{R}.\text{Query} \, |a\rangle \, |b\rangle \, |0\rangle = |a\rangle \, |b\rangle \, |\widetilde{R}_{ab}\rangle$$

  *with $(1 - \varepsilon)R_{ab} \leq \widetilde{R}_{ab} \leq (1 + \varepsilon)R_{ab}$ for every pair of vertices $a, b \in V$, and $R_{ab}$ being the effective resistance between vertices $a$ and $b$, in $\widetilde{O}(1/\varepsilon^2)$ time.*

*Proof.* First, we use the algorithm ClassicalEffectiveResistance$(G, \varepsilon)$ to obtain the matrix $Z_G$ and store each entries of matrix in QRAM. This allows us to access the matrix through a unitary $U_{Z_G}$ such that

$$U_{Z_G} \, |i\rangle \, |j\rangle \, |0\rangle = |i\rangle \, |j\rangle \, |Z_G(i, j)\rangle$$

where $Z_G(i, j)$ is the entry in the $i$-th row and $j$-th column of the matrix $Z_G$, $i \in [n]$, $j \in [q]$ with $q = \lceil 24 \log n/\varepsilon^2 \rceil$. The time of computing and storing $Z_G$ is $\widetilde{O}(m/\varepsilon^2)$, and each query of $U_{Z_G}$ has a time complexity of $\widetilde{O}(1)$. Consider the following procedure:

$$|i\rangle \, |j\rangle \, |0\rangle \, |0\rangle \, |0\rangle \xrightarrow{U'_{Z_G}} |i\rangle \, |j\rangle \, \Big(\bigotimes_{k=1}^{q} |Z_G(i, k)\rangle\Big)\Big(\bigotimes_{k=1}^{q} |Z_G(j, k)\rangle\Big) |0\rangle$$

$$\stackrel{\text{denote}}{=\joinrel=} |i\rangle \, |j\rangle \, |Z_G^i\rangle |Z_G^j\rangle \, |0\rangle$$

$$\xrightarrow{U_{\text{minus}}} |i\rangle \, |j\rangle \, |Z_G^i\rangle |Z_G^j\rangle |Z_G^i - Z_G^j\rangle$$

$$\xrightarrow{U_{\text{square}}} |i\rangle \, |j\rangle \, |Z_G^i\rangle |Z_G^j\rangle |\widetilde{R}_{ij}\rangle$$

$$\xrightarrow{U'_{Z_G}{}^\dagger} |i\rangle \, |j\rangle \, |0\rangle |0\rangle |\widetilde{R}_{ij}\rangle$$

where $Z_G^i$ is the $i$-th row of the matrix $Z_G$, and $U'_{Z_G}$ can be implemented using $O(q)$ queries of $U_{Z_G}$; $U_{\text{minus}}$ satisfies $U_{\text{minus}} \, |i\rangle \, |j\rangle \, |0\rangle = |i\rangle \, |j\rangle \, |i - j\rangle$, $U_{\text{square}}$ satisfies $U_{\text{square}} \, |i\rangle \, |0\rangle = |i\rangle \, |i^2\rangle$. This procedure uses $\widetilde{O}(1/\varepsilon^2)$ queries to $U_{Z_G}$ and $\widetilde{O}(1)$ additional arithmetic operations. Therefore, let $\mathcal{R}$.Query $= U'_{Z_G}{}^\dagger U_{\text{square}} U_{\text{minus}} U'_{Z_G}$, we confirm that this satisfies the requirements outlined in the proposition. $\qquad\square$

The following proposition formalizes the procedure of storing a graph in QRAM, which is made straightforward by the capabilities of QRAM.

**Proposition B.4** (Quantum Underlying Graph Storage). *Let $H = (V, E, w)$ is a hypergraph with vertex set $V$ of size $n$, edge set $E$ of size $m$, weights $w \in \mathbb{R}_{\geq 0}^E$. Suppose $G = (V, F, c)$ is a sparse underlying graph $G$ of $H$. There is a quantum data structure* UGraphStore, *that supports the following operations:*

- *Initialization:* UGraphStore$(G)$, *outputs an instance $\mathcal{G}$, in $\widetilde{O}(mr)$ time.*

- *Query:* $\mathcal{G}$.Query, *outputs a unitary satisfying*

$$\mathcal{G}.\text{Query} \, |e\rangle \, |f\rangle \, |0\rangle = |e\rangle \, |f\rangle \, |c_{e,f}\rangle$$

  *for every edge $(e, f) \in F$, where $f = \{i, j\} \in S_e$, and performs no action if $(e, f) \notin F$, in $\widetilde{O}(1)$ time. Here, we represent $|f\rangle$ as $|i\rangle \, |j\rangle$ for $f = \{i, j\}$.*

*Proof.* For the graph $G$ with edge set $F$ of size at most $m(r - 1)$, the initialization step is to store all the weights $c_{e,f}$ for the edges with indices $(e, f) \in F$ into an array using a QRAM of size $\widetilde{O}(mr)$ and in $\widetilde{O}(mr)$ QRAM classical write operations.

For the query operation, the $\mathcal{G}$.Query is the QRAM quantum query operation to the above array. $\qquad\square$

The following proposition concerns weight updates in the quantum overestimation algorithm.

**Proposition B.5** (Weight Computation for Overestimates). *Let $H = (V, E, w)$ be a hypergraph with vertex set $V$ of size $n$, edge set $E$ of size $m$, weights $\in \mathbb{R}^E_{\geq 0}$. Suppose $\mathcal{O}_H$ is a quantum oracle to $H$, $\mathcal{G}$ and $\mathcal{R}$ represent the instances of* UGraphStore *and* EffectiveResistance *for the sparse underlying graph $G = (V, F, c)$, respectively. Then, there exists a quantum algorithm* WeightCompute$(\mathcal{O}_H, \mathcal{G}, \mathcal{R})$, *such that*

$$\text{WeightCompute}(\mathcal{O}_H, \mathcal{G}, \mathcal{R}) \, |e\rangle \, |f\rangle \, |0\rangle = |e\rangle \, |f\rangle \, |c'_{e,f}\rangle$$

*where*

$$c'_{e,f} = \frac{c_{e,f} R_f}{\sum_{g \in S_e} c_{e,g} R_g} \cdot w_e, \tag{7}$$

*and $R_f$ is the query result of $\mathcal{R}$ on vertices of $f$. The algorithm requires $O(1)$ queries to $\mathcal{O}_H$, $O(r)$ queries to both $\mathcal{R}$ and $\mathcal{G}$, and runs in $\widetilde{O}(r)$ time.*

*Proof.* Consider the following procedure: for any $e \in E$ and $f \in S_e$, we have

$$|e\rangle \, |f\rangle \, |0\rangle \xmapsto{\mathcal{O}_H} |e\rangle \, |f\rangle \, |0\rangle \, (\otimes_{i \in e} |i\rangle) \, |0\rangle \, |w_e\rangle \, |0\rangle$$

$$\xmapsto{U_{\text{star}}} |e\rangle \, |f\rangle \, |0\rangle \, (\otimes_{g \in S_e} |g\rangle \, |0\rangle) \, |w_e\rangle \, |0\rangle$$

$$\xmapsto{\mathcal{G}.\text{Query}} |e\rangle \, |f\rangle \, |c_{e,f}\rangle \, |0\rangle \, (\otimes_{g \in S_e} |g\rangle \, |c_{e,g}\rangle \, |0\rangle) \, |w_e\rangle \, |0\rangle$$

$$\xmapsto{\mathcal{R}.\text{Query}} |e\rangle \, |f\rangle \, |c_{e,f}\rangle \, |R_f\rangle \, |0\rangle \, (\otimes_{g \in S_e} |g\rangle \, |c_{e,g}\rangle \, |R_g\rangle \, |0\rangle) \, |w_e\rangle \, |0\rangle$$

$$\xmapsto{U_{\text{mult}}} |e\rangle \, |f\rangle \, |c_{e,f}\rangle \, |R_f\rangle \, |w_e c_{e,f} R_f\rangle \, (\otimes_{g \in S_e} |g\rangle \, |c_{e,g}\rangle \, |R_g\rangle \, |c_{e,g} R_g\rangle) \, |w_e\rangle \, |0\rangle$$

$$\xmapsto{\mathcal{O}_H^\dagger, \mathcal{G}.\text{Query}^\dagger, \mathcal{R}.\text{Query}^\dagger} |e\rangle \, |f\rangle \, |w_e c_{e,f} R_f\rangle \, (\otimes_{g \in S_e} |c_{e,g} R_g\rangle) \, |0\rangle$$

$$\xmapsto{U_{\text{sum}}} |e\rangle \, |f\rangle \, |w_e c_{e,f} R_f\rangle \, (\otimes_{g \in S_e} |c_{e,g} R_g\rangle) \, |\Delta_e\rangle \, |0\rangle$$

$$\xmapsto{U_{\text{div}}} |e\rangle \, |f\rangle \, |w_e c_{e,f} R_f\rangle \, (\otimes_{g \in S_e} |c_{e,g} R_g\rangle) \, |\Delta_e\rangle \, |c'_{e,f}\rangle$$

where $\Delta_e$ represents the sum $\sum_{g \in S_e} c_{e,g} R_g$, $U_{\text{star}}$ satisfies $U_{\text{star}} (\otimes_{i \in e} |i\rangle) |0\rangle = \otimes_{g \in S_e} |g\rangle \, |0\rangle$, and $U_{\text{mult}}, U_{\text{sum}}, U_{\text{div}}$ denote basic arithmetic operations—multiplication, addition, and division, respectively, as previously. It's clear that this procedure meets the requirements stated in the proposition, since $|S_e| = O(r)$ for $\forall e \in E$. $\square$

**Proposition B.6** (Preparation for Overestimates). *Let $T \in \mathbb{N}$, $C \in \mathbb{R}$, $\varepsilon \in \mathbb{R}$, and $H = (V, E, w)$ be a hypergraph with rank $r$. Assume $\mathcal{O}_H$ is a quantum oracle to $H$. Suppose $\{\mathcal{G}^{(t)} : t \in [T]\}$ represents a sequence of instances of* UGraphStore *for the sparse underlying graphs $G^{(t)} = (V, F^{(t)}, c^{(t)})$ of $H$, and $\{\mathcal{R}^{(t)} : t \in [T]\}$ represents a sequence of instances of* EffectiveResistance *for the corresponding underlying graphs $G^{(t)}$ and $\varepsilon$. Then, there is a quantum data structure* QOverestimate, *that supports the following operations:*

- *Initialization:* QOverestimate$(\{\mathcal{G}^{(t)} : t \in [T]\}, \{\mathcal{R}^{(t)} : t \in [T]\}, \mathcal{O}_H, C, T)$, *outputs an instance $\mathcal{Z}$ in $\widetilde{O}(\sum_{t \in [T]} \tau_t)$ time, where $\tau_t$ denotes the needed time of both* UGraphStore$(G^{(t)})$ *and* EffectiveResistance$(G^{(t)}, \varepsilon)$.

- *Query: $\mathcal{Z}$.Query, outputs a unitary such that, for every $e \in E$*

$$\mathcal{Z}.\text{Query} \, |e\rangle \, |0\rangle = |e\rangle \, |z_e\rangle$$

*with*

$$z_e = C \cdot \frac{1}{T} \sum_{t \in [T]} \sum_{g \in S_e} \ell_{e,g}^{(t)}$$

*where $\ell_{e,g}^{(t)} = c_{e,g}^{(t)} R_g^{(t)}$, and $R_g^{(t)}$ is the query result of $\mathcal{R}^{(t)}$ on vertices of $g$. This query is executed in $\widetilde{O}(r \sum \iota_t)$ time, where $\iota_t$ represents the time required for querying both* UGraphStore *and* EffectiveResistance *for $G^{(t)}$.*

*Proof.* The case of initialization operation is straightforward. Aside from initializing for UGraphStore and EffectiveResistance, we store $C, T$ in QRAM in $\widetilde{O}(1)$ time, allowing access through a unitary $U_{C,T} : |i\rangle |0\rangle \rightarrow |i\rangle |C/T\rangle$. For the query operation, we consider the following procedure:

$$
|e\rangle |0\rangle \xmapsto{\mathcal{O}_H, U_{\text{star}}} |e\rangle \left( \otimes_{t=1}^T \left( \otimes_{g \in S_e} |g\rangle |0\rangle \right) \right) |0\rangle
$$

$$
\xmapsto{\mathcal{G}^{(t)}.\text{Query}, \mathcal{R}^{(t)}.\text{Query}} |e\rangle \left( \otimes_{t=1}^T \left( \otimes_{g \in S_e} |g\rangle |c_{e,g}^{(t)}\rangle |R_g^{(t)}\rangle |0\rangle \right) \right) |0\rangle
$$

$$
\xmapsto{U_{\text{mult}}} |e\rangle \left( \otimes_{t=1}^T \left( \otimes_{g \in S_e} |g\rangle |c_{e,g}^{(t)}\rangle |R_g^{(t)}\rangle |\ell_{e,g}^{(t)}\rangle \right) \right) |0\rangle
$$

$$
\xmapsto{U_{\text{sum}}} |e\rangle \left( \otimes_{t=1}^T \left( \otimes_{g \in S_e} |g\rangle |c_{e,g}^{(t)}\rangle |R_g^{(t)}\rangle |\ell_{e,g}^{(t)}\rangle \right) \right) | \sum_t \Delta_e^{(t)} \rangle |0\rangle
$$

$$
\xmapsto{U_{C,T}, U_{\text{mult}}} |e\rangle \left( \otimes_{t=1}^T \left( \otimes_{g \in S_e} |g\rangle |c_{e,g}^{(t)}\rangle |R_g^{(t)}\rangle |\ell_{e,g}^{(t)}\rangle \right) \right) | \sum_t \Delta_e^{(t)} \rangle |z_e\rangle
$$

where $\Delta_e^{(t)}$ represents the sum $\sum_{g \in S_e} c_{e,g}^{(t)} R_g^{(t)}$, and $U_{\text{clique}}, U_{\text{mult}}, U_{\text{sum}}$ denote basic arithmetic operations of clique generation, multiplication, and addition, respectively, as previously. The procedure can be executed in $O(r \sum_{t \in [T]} \iota_t)$ time, since $|S_e| = O(r)$ for $\forall e \in E$. $\qquad\square$

**Proposition B.7.** *Let $\mathcal{Z}$ be the output of* QHLSO$(\mathcal{O}_H, T, \alpha_1, \alpha_2)$ *(Algorithm 1). Then, the vector $z$ stored in $\mathcal{Z}$ is a $\nu$-overestimate for $H$, where $\nu = (1 + \alpha_2)C_1 n$ and $C_1$ is determined by $\alpha_1, \alpha_2, r, T$, as defined in line 8 of Algorithm 1.*

*Proof.* In the algorithm, $\widetilde{G}^{(t)}$ is a $\alpha_1$-spectral sparsifier of $G^{(t)}$. Let $R_f^{(\tilde{t})}$ and $R_f^{(t)}$ be the effective resistances of $\widetilde{G}^{(t)}$ and $G^{(t)}$ respectively. Since effective resistances correspond to quadratic forms in the pseudo-inverse of the Laplacian, we have $\frac{1}{1+\alpha_1} R_f^{(t)} \leq R_f^{(\tilde{t})} \leq \frac{1}{1-\alpha_1} R_f^{(t)}, \forall f \in F$. According to Proposition B.3, we know that $(1 - \alpha_2) R_f^{(\tilde{t})} \leq \widetilde{R}_f^{(t)} \leq (1 + \alpha_2) R_f^{(\tilde{t})}, \forall f \in F$. Combining two inequalities we obtain

$$
\frac{1 - \alpha_2}{1 + \alpha_1} \cdot R_f^{(t)} \leq \widetilde{R}_f^{(t)} \leq \frac{1 + \alpha_2}{1 - \alpha_1} \cdot R_f^{(t)}, \quad \forall f \in F.
$$

Let $\alpha_3 := \max\{1 - \frac{1-\alpha_2}{1+\alpha_1}, \frac{1+\alpha_2}{1-\alpha_1} - 1\} = \frac{\alpha_1+\alpha_2}{1-\alpha_1}$, then $\widetilde{R}_f^{(t)}$ is an $\alpha_3$-approximate of $R_f^{(t)}$ for $\forall f \in F$. We define $\widetilde{\ell}_{e,f}^{(t)} = \widetilde{c}_{e,f}^{(t)} \widetilde{R}_f^{(t)}$.

We will show that $z$ is a $\nu$-overestimate with corresponding underlying graph $G = (V, F, \bar{c})$, where $\bar{c} = \frac{1}{T} \sum_{t \in [T]} \widetilde{c}^{(t)}$. Specifically, we need to verify the following two conditions:

1. $\|z\|_1 \leq \nu$,

2. $z_e \geq w_e R_e$ for all $e \in E$ where $R_e = \max\{R_f : f \in \binom{e}{2}\}$.

We show the first condition first. As $\widetilde{\ell}_{e,f}^{(t)} = \widetilde{c}_{e,f}^{(t)} \widetilde{R}_f^{(t)} \leq (1 + \alpha_2) \widetilde{c}_{e,f}^{(t)} R_f^{(\tilde{t})}$, we have

$$
\|z\|_1 = \sum_{e \in E} C_1 \frac{1}{T} \sum_{t \in [T]} \sum_{g \in S_e} \widetilde{\ell}_{e,g}^{(t)} = C_1 \cdot \frac{1}{T} \sum_{t \in [T]} \left( \sum_{e \in E} \sum_{g \in S_e} \widetilde{\ell}_{e,g}^{(t)} \right) \leq C_1 (1 + \alpha_2) n
$$

where the final inequality is derived from Lemma A.1.

We now prove that the second condition also holds. For any $e \in E$ we fix $a \in e$. Since effective resistance is a metric on vertices, for any $u, v \in e$, it follows that

$$
R_{uv} \leq R_{ua} + R_{av}.
$$

Consequently, at least one of the two terms on the RHS must be at least $R_{uv}/2$. By taking the maximum on both sides, we obtain

$$
\max_{u,v \in e} R_{uv} \leq 2 \max_{u \in e} R_{au}
$$

Thus, for any $e \in E$, we have

$$
\begin{aligned}
\log\left(w_e R_e(\overline{c})\right) &\leq \log\left(w_e \cdot 2\max\{R_f(\overline{c}) : f \in S_e\}\right) \\
&\overset{\text{denote}}{=} \log(w_e \cdot 2R_{f^\star}(\overline{c})) \\
&\overset{(a)}{\leq} \frac{1}{T}\sum_{t\in[T]}\log\left(2w_e R_{f^\star}\left(c^{(t)}\right)\right) \\
&\leq \frac{1}{T}\sum_{t\in[T]}\log\left(2w_e(1+\alpha_3)\widetilde{R}_{f^\star}^{(t)}\right) = \frac{1}{T}\sum_{t\in[T]}\log\left(2(1+\alpha_3)w_e \cdot \widetilde{\ell}_{e,f^\star}^{(t)}/\widetilde{c}_{e,f^\star}^{(t)}\right) \\
&\overset{(b)}{=} \frac{1}{T}\sum_{t\in[T]}\left(\log\left(\frac{\widetilde{c}_{e,f^\star}^{(t+1)}\cdot\sum_{g\in S_e}\widetilde{\ell}_{e,g}^{(t)}}{\widetilde{c}_{e,f^\star}^{(t)}}\right)\right) + \log(2(1+\alpha_3)) \\
&= \frac{1}{T}\sum_{t\in[T]}\left(\log\left(\frac{\widetilde{c}_{e,f^\star}^{(t+1)}}{\widetilde{c}_{e,f^\star}^{(t)}}\right) + \log\left(\sum_{g\in S_e}\widetilde{\ell}_{e,g}^{(t)}\right)\right) + \log(2(1+\alpha_3)) \\
&\overset{(c)}{\leq} \frac{1}{T}\log\left(\frac{\widetilde{c}_{e,f^\star}^{(T+1)}}{\widetilde{c}_{e,f^\star}^{(1)}}\right) + \log\left(\frac{1}{T}\sum_{t\in[T]}\sum_{g\in S_e}\widetilde{\ell}_{e,g}^{(t)}\right) + \log(2(1+\alpha_3)) \\
&\overset{(d)}{\leq} \frac{1}{T}\log r + \log\left(\frac{1}{T}\sum_{t\in[T]}\sum_{g\in S_e}\widetilde{\ell}_{e,g}^{(t)}\right) + \log(2(1+\alpha_3)) \\
&\overset{(e)}{=} \log\left(z_e\right)
\end{aligned}
$$

where inequality $(a)$ holds since $\log\left(R_f(c)\right)$ is convex with respect to $c$ (see Lemma A.2), equality $(b)$ follows from definition of WeightCompute (Equation (7)), inequality $(c)$ follows from the concavity of $\log$, and inequality $(d)$ arises from the fact that

$$
\frac{\widetilde{c}_{e,f^\star}^{(T+1)}}{\widetilde{c}_{e,f^\star}^{(1)}} \leq \frac{w_e}{w_e/(|e|-1)} \leq r,
$$

the last equality $(e)$ follows directly from the definition of $z_e$ in Proposition B.6, with the parameter $C$ selected as in line 8 of Algorithm 1. $\qquad\square$

*Proof of Theorem 3.4.* By taking $\alpha_1 = \alpha_2 = 0.1$ as constants and $T = \log r$, $z$ becomes a $4n$-overestimate according to Proposition B.7. It remains to analyze the time complexity of the algorithm.

First, we note that the WeightInitialize$(\mathcal{O}_H)$ procedure runs in $\widetilde{O}(1)$ time, as established in Proposition B.1. In each round $t \in [T]$, GraphSparsify$(U_{G(t)}, \alpha_1)$ is executed in $\widetilde{O}(r\sqrt{mnr})$ time, following Theorem 2.6, where $\widetilde{O}(r)$ accounts for the query cost of $U_{G(t)}$. The resulting graph $\widetilde{G}^{(t)}$ is a $\alpha_1$-spectral sparsifier of $G^{(t)}$, and a crucial fact is that $\widetilde{G}^{(t)}$ is sparse and the number of edges in $\widetilde{G}^{(t)}$ is $\widetilde{O}(n)$. Hence, we can initialize the data structures UGraphStore$(\widetilde{G}^{(t)})$ and EffectiveResistance$(\widetilde{G}^{(t)}, \alpha_2)$ in $\widetilde{O}(n)$ time, as per Proposition B.3 and Proposition B.4. The procedure WeightCompute$(\mathcal{O}_H, \mathcal{R}^{(t)}, \mathcal{G}^{(t)})$ provides $U_{G(t+1)}$, with each query taking $\widetilde{O}(r)$ time according to Proposition B.5. In the final step of the algorithm, $\mathcal{Z}$ can be initialized in $\widetilde{O}(n)$ time and each $\mathcal{Z}$.Query can be executed in $\widetilde{O}(r)$ time, following Proposition B.6 with $\tau_t = \widetilde{O}(n)$ and $\iota_t = \widetilde{O}(1)$.

To summarize, the total preprocessing time is $\widetilde{O}(r\sqrt{mnr})$, and the per-query time is $\widetilde{O}(r)$. $\qquad\square$

## C. Proof of Theorem 4.1

The proof of correctness for the algorithm follows closely the chaining proofs in Lee (2023) and Jambulapati et al. (2023). In particular, we rely on the following crucial technical bound from Lee (2023), which is derived using Talagrand's generic chaining method.

**Lemma C.1** (Corollary 2.13 in Lee (2023)). *Let $A : \mathbb{R}^n \to \mathbb{R}^s$ be a linear map, with $a_1, \ldots, a_s$ representing the rows of $A$. The functions $\phi_1, \ldots, \phi_m : \mathbb{R}^s \to \mathbb{R}$ are in the form of $\phi_i(x) = \max_{j \in S_i} w_i |\langle a_j, x \rangle|$ for some $S_i \subseteq [s]$ and $w \in [0,1]^{S_i}$. Let $D = \max_{i \in [m]} |S_i|$. Then, for any $T \subseteq B_2^n := \{x \in \mathbb{R}^n : \|x\|^2 \leq 1\}$, the following inequality holds:*

$$\mathbb{E} \sup_{x \in T} \sum_{j=1}^m \xi_j \phi_j(x)^2 \leq C_0 \cdot \|A\|_{2 \to \infty} \sqrt{\log(s+n) \log D} \cdot \sup_{x \in T} \left( \sum_{j=1}^m \phi_j(x)^2 \right)^{1/2},$$

*for some universal constant $C_0$. The variables $\xi_1, \ldots, \xi_m$ are i.i.d. Bernoulli random variables taking values $\pm 1$, and $\|A\|_{2 \to \infty}$ is defined by $\|A\|_{2 \to \infty} := \max\{\|Ax\|_\infty : x \in B_2^n\}$.*

To provide clarity on how this lemma applies, we explain its connection to hypergraphs. Let $H = (V, E, w)$ be a hypergraph with $|E| = m, |V| = n$, and weights $w \in \mathbb{R}_{\geq 0}^E$. The rank of the hypergraph is $r = \max_{e \in E} |e|$. In the lemma, $n$ and $m$ correspond to the number of vertices and hyperedges, respectively. The functions $\phi_i$ capture the maximization over all edges in the clique generated by replacing each hyperedge, corresponding to the energy of the hyperedges. The number $s$ refers to the number of edges in the complete graph $K_n$, i.e., $s = n(n-1)/2$. The parameter $D$ represents the maximum number of edges in the clique generated by replacing each hyperedge, i.e., $D = r(r-1)/2$.

Before proving the Theorem 4.1, we present the following fact.

**Proposition C.2.** *Let $H = (V, E, w)$ be a hypergraph with $n$ vertices, and let $G = (V, F, c)$ represent its underlying graph. For any $x \in \mathbb{R}^n$ such that $x \perp 1$, the inequality $Q_H(L_G^{+/2} x) \geq \|x\|^2$ holds.*

*Proof.* For any $v \in \mathbb{R}^n$, we have

$$\begin{aligned}
Q_H(v) &= \sum_{e \in E} w_e \max_{\{i,j\} \subseteq e} (v_i - v_j)^2 \\
&= \sum_{e \in E} \Big( \sum_{\{i,j\} \subseteq e} c_{ij}^e \Big) \max_{\{i,j\} \subseteq e} (v_i - v_j)^2 \\
&\geq \sum_{e \in E} \sum_{\{i,j\} \subseteq e} c_{ij}^e (v_i - v_j)^2 \\
&= v^\top L_G v.
\end{aligned}$$

Taking $v = L_G^{+/2} x$ achieves we desired. $\square$

*Proof of Theorem 4.1.* Utilizing Algorithm 1 we can obtain a query access to $\nu$-overestimate $z$ with $\nu = 2(1 + \alpha_2)C_1 \cdot n = O(n)$ in $\widetilde{O}(r\sqrt{mnr})$ time. Each query can be executed in $\widetilde{O}(r)$ time. Combining with quantum sampling algorithm (Corollary 2.8), we can sample a subset $\widetilde{E} \subseteq E$ of size $M = \widetilde{O}(n/\varepsilon^2)$ in $\widetilde{O}(\sqrt{Mm} \cdot r) = \widetilde{O}(r\sqrt{mn}/\varepsilon)$ time. The quantum sum estimation (Theorem 2.9) is executed in $\widetilde{O}(r\sqrt{m})$ time, obtaining $\widetilde{s}$ as 0.1-approximation of $s = \|z\|_1$. We then move on to demonstrate the correctness of the algorithm, showing that the output $\widetilde{H}$ is indeed an $\varepsilon$-spectral sparsifier of $H$.

For a hyperegde $e \in E$, we introduce the following notations

$$\begin{aligned}
\mu_e &= z_e / \widetilde{s}, \\
a_{ij} &= L_G^{+/2} (\delta_i - \delta_j), \quad \forall \{i,j\} \in [n], \\
a_{ij}^e &= \sqrt{\widetilde{s} w_e / z_e} \cdot a_{ij}, \quad \forall \{i,j\} \subseteq e, \\
\phi_e(x) &= \max \left\{ |\langle a_{ij}^e, x \rangle| : \forall \{i,j\} \subseteq e \right\}, \quad \forall x \in \mathbb{R}^n, \\
\phi_e^2(x) &= \max \left\{ \langle a_{ij}^e, x \rangle^2 : \forall \{i,j\} \subseteq e \right\}, \quad \forall x \in \mathbb{R}^n,
\end{aligned}$$

where $R_e := \max\{R_f : f \in \binom{e}{2}\}$. Suppose we sample a hyperedge sequence $\sigma_\mu = (e_\mu^{(1)}, \ldots, e_\mu^{(M)})$ such that each element $e$ is sampled with probability proportional to $\mu_e$ (also proportional to $z_e$). For the new obtained hypergraph $H_\mu$, the weight of sampled hyperedge is

$$w_e^\mu = \frac{\# \left\{ t \in [M] : e_\mu^{(t)} = e \right\}}{M} \cdot \frac{w_e}{\mu_e}, \quad \forall e \in E.$$

Recall the definition that $Q_e(x) = \max_{\{u,v\} \subseteq e}(x_u - x_v)^2$, the energy of sampled hypergraph $H_\mu$ is given by

$$Q_{H_\mu} = \frac{1}{M} \sum_{t=1}^{M} \frac{w_{e_\mu^{(t)}}}{\mu_{e_\mu^{(t)}}} Q_{e_\mu^{(t)}}(x).$$

We want to choose sample time $M$ sufficiently large such that

$$\mathop{\mathbb{E}}_{H_\mu} \left[ \left| Q_H(x) - Q_{H_\mu}(x) \right| \right] \leq \varepsilon \cdot Q_H(x), \quad \forall x \in \mathbb{R}^n. \tag{8}$$

Equivalently, it suffices to show that

$$\mathop{\mathbb{E}}_{H_\mu} \max_{v: Q_H(v) \leq 1} \left| Q_H(v) - Q_{H_\mu}(v) \right| \leq \varepsilon.$$

It is worth noting that for $\forall x \in \mathbb{R}^n$,

$$\frac{w_e}{\mu_e} Q_e \left( L^{+/2} x \right) = \frac{w_e}{z_e} \cdot \widetilde{s} \max_{\{i,j\} \subseteq e} \left\langle L^{+/2} x, \delta_i - \delta_j \right\rangle^2 = \max_{\{i,j\} \subseteq e} \left\langle x, a_{ij}^e \right\rangle^2 = \phi_e^2(x).$$

Then we have

$$Q_{H_\mu}\left( L^{+/2} x \right) = \frac{1}{M} \sum_{j \in [M]} \phi_j^2(x) \tag{9}$$

where $\phi_j(x)$ corresponds to the $j$-th sampled hyperedge using $\mu$. Let $H_\mu'$ be an independent copy of $H_\mu$, and $\xi_t, t \in [M]$ be the i.i.d. Bernoulli $\pm 1$ random variables. Note that $\mathbb{E}_{H_\mu}\left[ Q_{H_\mu}(x) \right] = \frac{\widetilde{s}}{s} Q_H(x)$ and $\widetilde{s}/s \in [0.9, 1.1]$. By convexity of the absolute value function, for any $x \in T := \{ x \in \mathbb{R}^n : Q_H(L^{+/2}x) \leq 1 \}$, we have

$$\mathop{\mathbb{E}}_{H_\mu} \max_{x \in T} \left| Q_H\left( L^{+/2}x \right) - Q_{H_\mu}\left( L^{+/2}x \right) \right|$$

$$\overset{\text{concavity}}{\leq} \mathop{\mathbb{E}}_{H_\mu'} \mathop{\mathbb{E}}_{H_\mu} \max_{x \in T} \left| \frac{s}{\widetilde{s}} Q_{H_\mu'}\left( L^{+/2}x \right) - Q_{H_\mu}\left( L^{+/2}x \right) \right|$$

$$\leq \frac{s}{\widetilde{s}} \cdot \mathop{\mathbb{E}}_{H_\mu'} \mathop{\mathbb{E}}_{H_\mu} \max_{x \in T} \left| Q_{H_\mu'}\left( L^{+/2}x \right) - Q_{H_\mu}\left( L^{+/2}x \right) \right| + \left( \frac{s}{\widetilde{s}} - 1 \right) \mathop{\mathbb{E}}_{H_\mu} \max_{x \in T} \left| Q_{H_\mu}\left( L^{+/2}x \right) \right|$$

$$\overset{\text{Equation (9)}}{=} \frac{s}{\widetilde{s}} \cdot \mathop{\mathbb{E}}_{H_\mu'} \mathop{\mathbb{E}}_{H_\mu} \max_{x \in T} \left| \frac{1}{M} \sum_{t \in [M]} \phi_{e_\mu^{(t)}}^2(x) - \phi_{e_\mu^{(t)'}}^2(x) \right| + \left( \frac{s}{\widetilde{s}} - 1 \right) \mathop{\mathbb{E}}_{H_\mu} \max_{x \in T} \left| \frac{1}{M} \sum_{t \in [M]} \phi_{e_\mu^{(t)}}^2(x) \right|$$

$$= \frac{s}{\widetilde{s}} \cdot \mathop{\mathbb{E}}_{\xi} \mathop{\mathbb{E}}_{H_\mu'} \mathop{\mathbb{E}}_{H_\mu} \max_{x \in T} \left| \frac{1}{M} \sum_{t \in [M]} \xi_t \left( \phi_{e_\mu^{(t)}}^2(x) - \phi_{e_\mu^{(t)'}}^2(x) \right) \right| + \left( \frac{s}{\widetilde{s}} - 1 \right) \mathop{\mathbb{E}}_{\xi} \mathop{\mathbb{E}}_{H_\mu} \max_{x \in T} \left| \frac{1}{M} \sum_{t \in [M]} \xi_t \phi_{e_\mu^{(t)}}^2(x) \right|$$

$$\leq \frac{2s}{\widetilde{s}} \cdot \mathop{\mathbb{E}}_{\xi} \mathop{\mathbb{E}}_{H_\mu} \max_{x \in T} \left| \frac{1}{M} \sum_{t \in [M]} \xi_t \phi_{e_\mu^{(t)}}^2(x) \right| + \left( \frac{s}{\widetilde{s}} - 1 \right) \mathop{\mathbb{E}}_{\xi} \mathop{\mathbb{E}}_{H_\mu} \max_{x \in T} \left| \frac{1}{M} \sum_{t \in [M]} \xi_t \phi_{e_\mu^{(t)}}^2(x) \right|$$

$$= \left( \frac{3s}{\widetilde{s}} - 1 \right) \cdot \mathop{\mathbb{E}}_{\xi} \mathop{\mathbb{E}}_{H_\mu} \max_{x \in T} \left| \frac{1}{M} \sum_{t \in [M]} \xi_t \phi_{e_\mu^{(t)}}^2(x) \right|.$$

Note that for any function $f$, the inequality $\max_{x \in T} |f(x)| \leq \max\{\max_{x \in T} f(x), 0\} + \max\{\max_{x \in T} -f(x), 0\}$ holds. Now, let $f(x) = \frac{1}{M} \sum_{t \in [M]} \xi_t \phi_{e_\mu^{(t)}}^2(x)$. The second term $0$ in $\max\{\cdot, 0\}$, can be attained by the first term, since $\lim_{v \to 1} Q_H(v) = 0$, combined with the identity Equation (9). Consequently, we have

$$\mathop{\mathbb{E}}_{H_\mu} \max_{x \in T} \left| Q_H\left( L^{+/2}x \right) - Q_{H_\mu}\left( L^{+/2}x \right) \right|$$

$$\leq \left( \frac{3s}{\widetilde{s}} - 1 \right) \cdot \left( \mathop{\mathbb{E}}_{\xi} \mathop{\mathbb{E}}_{H_\mu} \max_{x \in T} f(x) + \mathop{\mathbb{E}}_{\xi} \mathop{\mathbb{E}}_{H_\mu} \max_{x \in T} -f(x) \right) \tag{10}$$

$$= 2\left( \frac{3s}{\widetilde{s}} - 1 \right) \cdot \mathop{\mathbb{E}}_{\xi} \mathop{\mathbb{E}}_{H_\mu} \max_{x \in T} \frac{1}{M} \sum_{t \in [M]} \xi_t \phi_{e_\mu^{(t)}}^2(x).$$

The last equality holds because $\xi_t, t \in [T]$ are i.i.d. Bernoulli $\pm 1$ random variables.

Consider the random process $V_x = \frac{1}{M} \sum_{j \in [M]} \xi_j \phi_j^2(x)$, where $\phi_j(x)$ corresponds to the $j$-th sampled hyperedge using $\mu$, $x \in T$. By applying Lemma C.1 with the linear map $A : \mathbb{R}^n \to \mathbb{R}^{n(n-1)/2}$ defined as

$$(Ax)_{ij} := \max_{e \in E : \{i,j\} \in \binom{e}{2}} \|a_{ij}^e\| \cdot \langle x, a_{ij} / \|a_{ij}\| \rangle,$$

and using Proposition C.2 to ensure that $T \subseteq B_2^n$, as required by Lemma C.1, we obtain:

$$
\begin{aligned}
\mathbb{E} \sup_{\xi \ x \in T} V_x &\le C_0 \cdot \frac{\|A\|_{2 \to \infty} \cdot \sqrt{\log(n(n-1)/2 + n) \log(r(r-1)/2)}}{\sqrt{M}} \max_{x \in T} \left( \frac{1}{M} \sum_{j \in [M]} \phi_j^2(x) \right)^{1/2} \\
&\le 2C_0 \cdot \frac{\|A\|_{2 \to \infty} \cdot \sqrt{\log n \log r}}{\sqrt{M}} \max_{x \in T} \left( \frac{1}{M} \sum_{j \in [M]} \phi_j^2(x) \right)^{1/2}
\end{aligned}
\tag{11}
$$

where

$$
\begin{aligned}
\|A\|_{2 \to \infty} &= \max\{\|Ax\|_\infty : x \in B_2^n\} = \max_{e \in E} \max_{\{i,j\} \subseteq e} \|a_{ij}^e\| \\
&= \max_{e \in E} \max_{\{i,j\} \subseteq e} \sqrt{\frac{\widetilde{s} w_e}{z_e} \cdot R_{ij}} = \sqrt{\widetilde{s}} \cdot \max_{e \in E} \max_{\{i,j\} \subseteq e} \sqrt{\frac{w_e R_{ij}}{z_e}} \\
&\le \sqrt{\widetilde{s}}.
\end{aligned}
$$

Observe that the component in RHS of Equation (11) can be written as

$$\max_{x \in T} \frac{1}{M} \sum_{j \in [M]} \phi_j^2(x) = \max_{v : Q_H(v) \le 1} \frac{1}{M} \sum_{j \in [M]} \phi_j^2 \left( L^{+/2} v \right) = \max_{v : Q_H(v) \le 1} Q_{H_\mu}(v).$$

The first equality holds because $Q_H(x) = Q_H(x')$ whenever $x - x' \in \ker(L_G)$. Then we have

$$
\begin{aligned}
\tau := \mathbb{E}_{H_\mu} \max_{v : Q_H(v) \le 1} \left| Q_H(v) - Q_{H_\mu}(v) \right| &= \mathbb{E}_{H_\mu} \max_{x \in T} \left| Q_H \left( L^{+/2} x \right) - Q_{H_\mu} \left( L^{+/2} x \right) \right| \\
&\overset{\text{Equation (10)}}{\le} \left( \frac{6s}{\widetilde{s}} - 2 \right) \cdot \mathbb{E}_{\varepsilon} \mathbb{E}_{H_\mu} \max_{x \in T} \frac{1}{M} \sum_{t \in [M]} \varepsilon_t \phi_{e_\mu^{(t)}}^2(x) \\
&\overset{\text{Equation (11)}}{\le} 4 \left( \frac{3s}{\widetilde{s}} - 1 \right) \cdot C_0 \cdot \sqrt{\widetilde{s} \log n \log r / M} \cdot \mathbb{E}_{H_\mu} \left( \max_{v : Q_H(v) \le 1} Q_{H_\mu}(v) \right)^{1/2} \\
&\overset{\text{concavity}}{\le} 4 \left( \frac{3s}{\widetilde{s}} - 1 \right) \cdot C_0 \cdot \sqrt{\widetilde{s} \log n \log r / M} \cdot \left( \mathbb{E}_{H_\mu} \max_{v : Q_H(v) \le 1} Q_{H_\mu}(v) \right)^{1/2} \\
&\le 4 \left( \frac{3s}{\widetilde{s}} - 1 \right) \cdot C_0 \cdot \sqrt{\widetilde{s} \log n \log r / M} \cdot (1 + \tau)^{1/2} \\
&\le 4 \left( \frac{3s}{\widetilde{s}} - 1 \right) \cdot C_0 \cdot \sqrt{\widetilde{s} \log n \log r / M} \cdot \left( 1 + \frac{1}{2}\tau \right) \\
&\le 10 C_0 \cdot \sqrt{\widetilde{s} \log n \log r / M} \cdot \left( 1 + \frac{1}{2}\tau \right).
\end{aligned}
$$

Therefore, we have $\tau \le 20 C_0 \sqrt{\widetilde{s} \log n \log r / M}$ whenever $M \ge 100 C_0^2 \widetilde{s} \log n \log r$. Choosing $M := 400 C_0^2 \widetilde{s} \log n \log r / \varepsilon^2 = \Theta(n \log n \log r / \varepsilon^2)$ yields

$$\mathbb{E}_{H_\mu} \max_{v : Q_H(v) \le 1} \left| Q_H(v) - Q_{H_\mu}(v) \right| = \tau \le \varepsilon.$$

$\square$

