# OpenReview forum: "Quantum Speedup for Hypergraph Sparsification"
_ICML.cc/2025/Conference — ICML 2025 poster_

### Official Review · Reviewer_vTAd · 2025-03-10

**Overall Recommendation:** 4

**Summary:**

Graph sparsification has been extensively studied [SS11, BSS12, LS17] and has numerous applications in graph algorithms and machine learning. As a natural generalization of graphs, hypergraphs have gained increasing attention. Similarly, hypergraph sparsification has attracted significant interest following the pioneering work of [SY'19]. Motivated by the successful application of quantum computing to graph sparsification [AD'20], this paper presents the first quantum algorithm for hypergraph sparsification. Specifically, for a hypergraph $H$ with $n$ vertices and $m$ edges, the proposed algorithm constructs an $\epsilon$-spectral sparsifier of size $O(n \log n \log r / \epsilon^2)$ in time $\widetilde{O}(r \sqrt{mn} / \epsilon)$, where $r$ denotes the rank. This result significantly outperforms the best known sequential algorithm, which runs in $\widetilde{O}(mr)$ time [JLS'23].
Moreover, the proposed quantum algorithm extends naturally to quantum hypergraph cut sparsification, mincut solving, and $s-t$ mincut solving, broadening its applicability to fundamental problems in hypergraph optimization.

$\textbf{Reviewer vTAd update after rebuttal:}$ I thank the authors for their clear clarification. I will retain my score and am inclined to recommend acceptance of this paper.

**Claims And Evidence:**

The main theorem 4.3 and three corollaries 5.1, 5.2, and 5.3 are clearly stated and proved.

**Essential References Not Discussed:**

(1) $\textbf{Cut Sparsification and Succinct Representation of Submodular Hypergraphs}$, ICALP 2024.
This paper explored the cut sparsifier of submodular hypergraphs.
(2) $\textbf{Near-optimal Linear Sketches and Fully-Dynamic Algorithms for Hypergraph Spectral Sparsification}$, STOC 2025.
This paper proposed algorithms for hypergraph spectral sparsifier under the fully-dynamic settings, which allow hyperedge insertions/deletions.

**Experimental Designs Or Analyses:**

No experiments.

**Methods And Evaluation Criteria:**

This paper is purely theoretical and has no experiments.

**Other Comments Or Suggestions:**

(1) In the paper "Hypergraph Diffusions and Resolvents for Norm-Based Hypergraph Laplacians", Ameranis et al. proposed the first nearly-linear-time algorithm for approximately computing resolvents of the hypergraph Laplacian operator. An intriguing direction for future research could be exploring quantum speedup techniques to further improve its running time.
(2) Typo: Line 100, adopts -> adopt

**Other Strengths And Weaknesses:**

$\textbf{Strengths:}$ (1) Hypergraph sparsification has various applications and has been extensively studied in the past few years. This paper proposed the first quantum algorithm for hypergraph sparsification.
(2) The proposed quantum hypergraph sparsifier has nearly linear size $\widetilde{O}(n / \epsilon^2)$ and takes time $\widetilde{O}(r \sqrt{mn} / \epsilon)$, which outperforms the running time $\widetilde{O}(m r)$ of [JLS'23, Lee'23], under the settings $\epsilon \ge \sqrt{n/m}$ and $m \ge n r$. When $r$ is a constant, this time complexity almost matches the lower bound $\Omega(m)$.
(3) The motivation and idea of this paper are straightforward and natural. Additionally, this paper is well-written and easy to understand.
$\textbf{Weaknesses:}$ (1) The running time of the proposed quantum algorithm improves the classical complexity $\widetilde{O}(m r)$ under the two assumptions $m \ge n r$ and $\epsilon \ge \sqrt{n / m}$, which weakens this paper's contribution.
(2) This paper follows the sampling-based framework of [JLS'23] and primarily builds on existing techniques from [AD'20, Hamoudi'22], which somewhat limits its novelty.

**Questions For Authors:**

See Weaknesses.

**Relation To Broader Scientific Literature:**

This paper outlines several potential directions for future work. Given the wide range of applications of graph and hypergraph sparsification, the proposed quantum graph/hypergraph sparsifier algorithm naturally opens the door to developing quantum algorithms for other graph-related problems. Additionally, considering existing research on directed and online hypergraph sparsification, it would be interesting to explore quantum algorithms for these settings as well.

**Theoretical Claims:**

I reviewed the proofs in the supplementary material but didn't read them carefully.

---

> ### Author Rebuttal · Authors · 2025-03-31
>
> We sincerely thank the reviewer for their thorough evaluation and
> constructive feedback. Below, we address the key concerns raised:
>
> 1. Essential References Not Discussed:
>
> Thanks for point out these two references, we will add them for the
> revision. Thanks for pointing out these references. We will add them in
> the revised version of our paper.
>
> 2. Running Time Assumptions:
>
> Regarding the assumptions ($\varepsilon>\sqrt{n / m}$ and $m>n r$), we
> would like to point out that they are reasonable and arise naturally in
> the context of the sparsification task: (a) $\varepsilon>\sqrt{n/m}$ is
> a natural assumption in the sparsification task, as this is equivalent
> to that the resulting sparsified graph (with $O(n/\varepsilon^2)$ edges)
> contains less edges than the original graph ($O(m)$ edges). This
> assumption also appears in previous works quantum sparsification
> algorithms (e.g., [AdW'20]). (b) $m>n r$ naturally holds whenever
> hypergraphs are not highly sparse. In dense hypergraphs, the number of
> hyperedges scales as $m=\Theta\left(n^r\right) \gg n r$. In practice,
> $r$ is typically treated as a constant greater than 2, which means that
> the number of hyperedges only needs to be larger than the linear size of
> the number of vertices.
>
> 3. Novelty and Technical Contributions:
>
> We acknowledge that our algorithmic analysis builds on the results of
> [JLS'23]. However, our core subroutine, QHLSO, is inspired by another
> critical work [Jambulapati et al.'2023]. The non-trivial contribution
> lies in identifying, adapting, and synthesizing existing classical
> frameworks to the quantum setting---a task requiring meticulous
> integration of recent classical and quantum algorithmic tools, including
> [AdW'20, Hamoudi'22]. The classical literature on hypergraph
> sparsification encompasses numerous advanced approaches, and selecting
> the right framework for quantum acceleration demanded substantial
> domain-specific insight. Furthermore, we intentionally prioritized
> readability to provide the quantum algorithms community with a clear
> foundation for exploring broader applications in this domain, while
> demonstrating how classical and quantum techniques can be cohesively
> combined to achieve novel efficiencies.
>
> 4. Future Work Suggestion:
>
> We thank the reviewer for directing us to the resolvent computation work
> by Ameranis et al. Exploring quantum speedups for hypergraph diffusion
> is a compelling direction, and we will mention this in our revised
> future work section.
>
> 5. Typos and Grammar:
>
> We will meticulously proofread the manuscript to address grammatical
> errors, including the noted typo (Line 100: "adopts" → "adopt").

---

### Official Review · Reviewer_fTGG · 2025-03-12

**Overall Recommendation:** 3

**Summary:**

The authors claim to give the first quantum algorithm for hypergraph sparsification. Their main theorem claims that they can find a sparsifier of size $O(n/\epsilon^2)$ in time $O(r \sqrt{mnr} + r\sqrt{mn}/ \epsilon)$ with high probability. Besides the introduction, the paper is concerned with proving this result.

**Claims And Evidence:**

All theorems and claims are supported with proofs. I am unable to verify that the proofs are correct.

**Essential References Not Discussed:**

I am unfamiliar with the literature on quantum algorithms, so I am not sure if any references were missed.

**Experimental Designs Or Analyses:**

The paper does not contain any experiments.

**Methods And Evaluation Criteria:**

The paper does not contain any experiments, or other evaluation methods.

**Other Comments Or Suggestions:**

None

**Other Strengths And Weaknesses:**

Unsure

**Questions For Authors:**

None

**Relation To Broader Scientific Literature:**

I am not an expert in quantum algorithms, however, the fact that this algorithm can provide sublinear running times in dense hypergraphs is of interest, as in the classical setting this seems like an unlikely result.

**Theoretical Claims:**

I have read the proofs and skimmed the appendix, and the claims look plausible, but since I lack expertise in quantum algorithms I can't judge the correctness very well.

---

> ### Author Rebuttal · Authors · 2025-03-31
>
> Thank you for your comments and review. Feel free to reach out if
> additional clarifications are needed.

---

### Official Review · Reviewer_EHoy · 2025-03-12

**Overall Recommendation:** 3

**Summary:**

This work introduces the first quantum algorithm for hypergraph sparsification, producing an $\varepsilon$-spectral sparsifier of size $\widetilde{O}(n / \varepsilon^2)$ in time $\widetilde{O}(r \sqrt{m n} / \varepsilon)$ for a weighted hypergraph with $n$ vertices, $m$ hyperedges, and rank $r$. This result demonstrates a quantum speedup over the classical state-of-the-art $\widetilde{O}(m r)$-time algorithm (Jambulapati et al., 2023; Lee, 2023) and matches the quantum lower bound of $\widetilde{\Omega}(\sqrt{m n} / \varepsilon)$ for $r = 2$ (Apers & de Wolf, 2020). The method combines a classical sampling-based framework with quantum techniques, including quantum graph sparsification (Apers & de Wolf, 2020), state preparation (Hamoudi, 2022), and sum estimation. Applications include sublinear-time quantum algorithms for computing hypergraph cut sparsifiers and approximating hypergraph mincuts and $s$-$t$ mincuts.

The three primary contributions align with Sections 3, 4, and 5, summarized as follows.

**Quantum Algorithm for Leverage Score Overestimate (Section 3)**

The authors introduce a quantum algorithm (Algorithm 1) to estimate hyperedge leverage score overestimates, based on classical approaches from Cohen et al. (2019) and Jambulapati et al. (2023). They adapt the previous concept of group leverage score overestimate and give a slightly different algorithm. The algorithm iteratively updates the weights $c^{(t)}$ of the underlying graph (line 2 of Algorithm 1), sparsifying the underlying graph (GraphSparsify), and then letting $c^{(t+1)}$ take the weight of the hyperedge with the proportion of its energy (WeightCompute). Then, it outputs the average of energies $\ell^{(t)}$ with appropriate scaling (QOverestimate). Of course, this algorithm is in a quantum way, and specially, the authors describe their subroutines and output as quantum data structures with initialization and query capabilities. The overestimate property (Proposition B.7) is proven using a telescoping argument akin to Cohen et al. (2019). The time complexity of this algorithm hinges on the graph sparsification step, and the quantum speedup here is mainly achieved through the result of Apers & de Wolf (2020).

**Quantum Hypergraph Sparsification (Section 4)**

Algorithm 2 presents a quantum sampling approach for hypergraph sparsification. It leverages the MultiSample subroutine to access the leverage score overestimate vector and samples a sequence of hyperedges with probabilities proportional to their overestimates. By combining the information of each sampled hyperedge with the normalization factor obtained via SumEstimate, the algorithm then reweights the hypergraph. Correctness is established via a chaining argument, adapted from Lee (2023) and Jambulapati et al. (2023). The time complexity is dominated by the sampling phase, which benefits from the precomputed overestimate data structure from Algorithm 1 and the quantum sampling subroutine from Corollary 2.8.

**Applications (Section 5)**

As a direct application, a cut sparsifier for hypergraphs is obtained. Since the hypergraph cut sparsifier preserves the cut energy, the quantum speedup for hypergraph sparsification extends to mincut and $s$-$t$ mincut problems.

**Claims And Evidence:**

The primary claim—that the algorithm constructs an $\varepsilon$-spectral sparsifier in $\widetilde{O}(r \sqrt{m n} / \varepsilon)$ time—is substantiated by Theorem 1.1 (formalized as Theorem 4.1). A detailed proof, provided in Appendices, employs leverage score overestimates and adapts a chaining argument from Lee (2023) to establish correctness. The speedup over the classical $\widetilde{O}(m r)$ bound is evident under reasonable assumption $\varepsilon \ge \sqrt{n/ m}$. To achieve the speedup, the authors design many quantum subroutines (e.g., GraphSparsify, MultiSample), which are derived using known techniques such as quantum graph sparsification (Apers & de Wolf, 2020) and basic operations such as addition and multiplication. Applications outlined in Section 5 are straightforward to see, though details are not fully provided. No other unsupported claims stand out; all the evidence is theoretical analysis.

**Essential References Not Discussed:**

All essential references appear to be appropriately discussed.

**Experimental Designs Or Analyses:**

NA

**Methods And Evaluation Criteria:**

The methods presented make sense for the challenges of hypergraph sparsification. The algorithm is built on a sampling-based framework, utilizing hyperedge leverage score overestimates derived by quantum graph sparsification (Apers & de Wolf, 2020) and some calculations. Other quantum techniques are employed, including preparing multiple state copies (Hamoudi, 2022) for sampling and sum estimation (Li et al., 2019) for reweighting. The evaluation centers on theoretical time complexity, the number of quantum gates, queries and QRAM operations. This is an increasingly adopted way to evaluate quantum algorithms (e.g., Apers & de Wolf, 2020). No benchmark datasets are used, as expected for a theoretical contribution.

**Other Comments Or Suggestions:**

Some typos and minor comments are listed below:

1. In Section 2, line 190 (left), notation $D$ is redundant; line 210 (right), 'an weighted' -> 'a weighted';
2. In Section 3, line 308 (left), 'a underlying' -> 'an underlying'; line 329 (left), 'a underlying' -> 'the underlying'; line 304 (right), $c_{e,f}$;
3. In Section 4, line 347 (right), 'denote by' -> 'denoted by'; line 348 (right), $|\tilde{E}| = O(n \log n \log r / \varepsilon^2)$;
4. In Section 5, line 390 (left), 'directly corollary' -> 'direct corollary'; line 410 (left), 'sparsity the' -> 'sparsify the';
5. In Section 6, line 392 (right), $O$ -> $\widetilde{O}$; line 429 (right), 'whether we' is redundant; line 437 (right), 'none which' -> 'none of which';
6. In Appendix, line 617, lack an 'is'; line 793, 'corresponds' -> 'correspond'; line 816, lack an 'is'; line 887, $\xi_1, \ldots, \xi_M$ -> $\xi_1, \ldots, \xi_m$;
7. In Proposition B.5, the time seems to be $\widetilde{O}(r/\varepsilon^2)$, since each query to $\mathcal{R}$ requires $\widetilde{O}(1/\varepsilon^2)$ time from Proposition B.3;
8. In Proposition B.6, the first step seems to use $U_{\mathrm{star}}$ instead of $U_{\mathrm{clique}}$, as in Proposition B.5;
9. In line 1047, I suggest additional justification for the first inequality.

**Other Strengths And Weaknesses:**

Strengths:

- The paper presents the first quantum algorithm for the fundamental problem of hypergraph sparsification.

Weakness:

- The techniques employed in this paper are largely adapted from previously established methods (e.g., Lee, 2023; Jambulapati et al., 2023; Apers & de Wolf, 2020), and the technical contribution appears to be somewhat incremental.

- About the write-up: The main content of the submission does not provide much detail on the formal techniques, instead devoting too much space to preliminaries. I suggest that the authors expand the later sections with more substantive information and include some proofs to help readers better understand and verify the claims.

**Questions For Authors:**

- I have provided some comments in the *Theoretical Claims* section above. It would be great if the authors could address them.

- Furthermore, could you briefly explain why the time complexity has a linear dependency on  $r$? What are the main barriers to improving this dependency in your current approach?

**Relation To Broader Scientific Literature:**

The paper builds on and extends several areas of prior literature:

- Graph Sparsification: It utilizes the result of quantum graph sparsification (Apers & de Wolf, 2020) and generalizes the graph case to hypergraphs, addressing future research directions outlined in Apers & de Wolf (2020).

- Hypergraph Sparsification: By incorporating quantum speedups, it advances classical algorithms (Lee, 2023; Jambulapati et al., 2023), achieving sublinear time complexity while preserving near-linear size.

- Quantum Algorithms: It leverages and extends quantum tools like state preparation (Hamoudi, 2022) and sum estimation (Li et al., 2019) to achieve the speedup.

**Theoretical Claims:**

I almost reviewed the entire theoretical claims in this paper, including the proofs in Appendices. I think the following needs to be addressed.

1. **Unitary Operations:** It would be helpful to include brief explanations of the unitary properties of some basic operations (e.g., $U_{\mathrm{div}}, U_{\mathrm{square}}, U_{\mathrm{star}}$).

2. **Initialization of EffectiveResistance in Proposition B.3:** This part is quite confusing.

- The authors attribute their approach to Claim 7.9 from Apers & de Wolf (2020) (abbreviated as the AW paper). However, Claim 7.9 in the AW paper does not mention or use the $ \tilde{O}(m/\epsilon^2)$  time complexity stated in Proposition B.3.

- Moreover, the AW paper actually provides a quantum method for obtaining $Z_G$  with a better time complexity, specifically $\widetilde{O}(\sqrt{mn}/\varepsilon + n/\varepsilon^4)$. Despite this, the authors opt for the slower classical algorithm with  $\tilde{O}(m/\epsilon^2)$ time complexity instead of leveraging the more efficient quantum approach.

- Thus, the authors should either justify their preference for the classical method or consider adopting the faster quantum alternative from the AW paper.

3. **Appendix C:** In line 940, you want to prove the claim that : $E_{H_{\mu}} [ | Q_H(x) - Q_{H_{\mu}}(x) | ] \leq \epsilon \cdot Q_H(x), \quad \forall x \in \mathbb{R}^n$. Later, you proved in Line 1047 that $E_{H_{\mu}} \max_{x: \|x\| \leq 1} | Q_H(x) - Q_{H_\mu}(x) | \leq \epsilon$. How do you use this result to complete the proof of the original claim?

---

> ### Author Rebuttal · Authors · 2025-03-31
>
> We sincerely appreciate the reviewer’s thorough evaluation and constructive feedback. Below, we address the key concerns raised:
>
> 1. Unitary Operations:
>
> The unitary operators $U_{\mathsf{mult}},U_{\mathsf{sum}},U_{\mathsf{div}},U_{\mathsf{square}},U_{\mathsf{minus}}$  are
> quantum gate implementation of the basic
> arithmetic operations. Specifically,
> they satisfy:
> $U_{\mathsf{mult}}\ket{a}\ket{b}\ket{0} = \ket{a}\ket{b}\ket{ab}$,
> $U_{\mathsf{sum}}\ket{a}\ket{b}\ket{0} = \ket{a}\ket{b}\ket{a+b}$,
> $U_{\mathsf{div}}\ket{a}\ket{b}\ket{0} = \ket{a}\ket{b}\ket{a/b}$,
> $U_{\mathsf{square}}\ket{a}\ket{0} = \ket{a}\ket{a^2}$,
> $U_{\mathsf{minus}}\ket{a}\ket{b}\ket{0} = \ket{a}\ket{b}\ket{a-b}$.
> And $U_{\textup{star}}$ satisfies $U_{\textup{star}}(\otimes_{i \in e}\ket{i})\ket{0}=\otimes_{g \in S_e}\ket{g}\ket{0}$, as defined in line 746.
> These unitary transformations offer quantum counterparts to classical operations while preserving computational efficiency.
>
> 2. Initialization of EffectiveResistance (Proposition B.3):
>
> We sincerely appreciate your careful reading and would like to clarify the technical details here.  In Claim 7.9 of the AW paper, the stated runtime $\widetilde{O}\left(\sqrt{m n} / \varepsilon+n / \varepsilon^4\right)$ comprises two components:
> (a). The first term $\widetilde{O}(\sqrt{m n} / \varepsilon)$: This corresponds to the time for quantum sparsification, which produces a sparse graph with $m^{\prime}=\widetilde{O}\left(n / \varepsilon^2\right)$ hyperedges.
> (b). The second term $\widetilde{O}\left(n / \varepsilon^4\right)$: This arises from applying the classical algorithm (Theorem B.2 and Theorem B.3 in our paper) to compute effective resistances on the sparsifier. Specifically, this step incurs a cost of $\widetilde{O}\left(m^{\prime} / \varepsilon^2\right)=$ $\widetilde{O}\left(n / \varepsilon^4\right)$.
> Our processing is consistent with that in the AW paper, but for the sake of algorithm clarity, we carefully write out the implementation of each step.
>
> 3. Appendix C:
>
> We acknowledge the need for greater clarity and will revise the text to explicitly outline the equivalence between the original claim (line 940)
> $$
> E_{H_\mu}\left[\left|Q_H(x)-Q_{H_\mu}(x)\right|\right] \leq \varepsilon \cdot Q_H(x), \quad \forall x \in {R}^n,
> $$
> and the proved result (line  1043)
> $$
> E_{H_\mu} \max_{x: Q_H(x) \leq 1}   \left|Q_H(x)-Q_{H_\mu}(x)\right| \leq \varepsilon
> $$
> More specifically, for any $x \perp 1$, we have  $Q_H(x)>0$. We then define $z=x / \sqrt{Q_H(x)}$. By homogeneity of $Q_H$, this ensures $Q_H(z)=1$, placing $z$ in the set $T=\lbrace x: Q_H(x) \leq 1\rbrace$. The result in Line 1043 implies that for all $z\in T$,
> $$
> E_{H_\mu}\left[\left|Q_H(z)-Q_{H_\mu}(z)\right|\right] \leq \varepsilon.
> $$
> Substituting $z=x / \sqrt{Q_H(x)}$ gives:
> $$
> E_{H_\mu}\left[\left|\frac{Q_H(x)-Q_{H_\mu}(x)}{Q_H(x)}\right|\right] \leq \varepsilon.
> $$
> Multiplying through by $Q_H(x)$ yields the original claim.
>
> 4. Technical Contributions:
>
> We agree that our algorithms builds on the previous results
>  (e.g., Lee, 2023; Jambulapati et al.,
> 2023; Apers de Wolf, 2020). The non-trivial
> contribution lies in identifying, adapting, and synthesizing
> existing classical frameworks to the quantum setting---a task
> requiring meticulous integration of recent classical and quantum
> algorithmic tools. The classical
> literature on hypergraph sparsification encompasses numerous
> advanced approaches, and selecting the right framework for quantum
> acceleration demanded substantial domain-specific insight.
>
> 5. About the write-up:
>
> We appreciate the reviewer’s feedback on the balance between the introduction and formal techniques. We will carefully revise the manuscript to streamline the preliminaries and move some important parts from the appendix to the main text, thereby enhancing clarity and facilitating the verification of our claims.
>
>
> 6. Typos and Grammar:
>
> We sincerely appreciate the reviewers’ careful reading and for pointing out these typos. We will meticulously proofread the manuscript to correct grammatical errors, including the noted typos.
>
> 7. Dependency on rank:
>
> The linear dependency on $r$ arises because our quantum algorithm QHLSO converts each hyperedge into a star graph (with $O(r)$ edges) to construct the underlying Laplacian system. This step inherently scales linearly with $r$, as each hyperedge of size $r$ requires explicit interactions among its vertices.
> Notably, our algorithm's linear dependence on $r$ is already an improvement over the quadratic dependency $O(r^2)$, which is achieved by utilizing a sparse underlying graph (see Definition 3.3) instead of general underlying graph (see  Definition 3.1).
> Further reducing this dependency is challenging: classical hypergraph sparsification methods face fundamental limits, and quantum representations inherently require enumerating all $r$ vertices in a hyperedge for unitary operations. Thus, both classical and quantum approaches encounter a barrier for hyperedge processing.

---

### Official Review · Reviewer_Kw2T · 2025-03-13

**Overall Recommendation:** 4

**Summary:**

Hypergraph sparsification is the process of reducing the number of hyper edges of a graph while preserving (as much as possible) the energy of the graph.

The paper introduces an algorithm for hypergraph sparsification, addressing an open problem proposed in a previous paper by Apers and de Wolf. More specifically, the authros show that  give an hypergraph with n vertices, m hyper edges and rank r, and error parameter e, e-eparsifier with Õ(n/e^2) edges, in time Õ(r*sqrt(mn)/e).

When rank r is constant, the proposed algorithm matches the quantum lower bound. Additionally, it provides a quantum speedup with respect to state of the art classical algorithms, which run in time Õ(mr).

To obtain the results, the authors obtain a faster quantum algorithm to compute the hyperedge leverage score overestimate, providing a quantum speedup over a classical algorithm proposed by Lee and Jambulapati (2023). They also use of a technique introduced by Hamoudi to make copies of a quantum state specified by an oracle.

**Claims And Evidence:**

Yes, the proofs of the claims are provided in the appendix, although I did not check carefully all proofs.

**Essential References Not Discussed:**

Not that I’m aware of.

**Experimental Designs Or Analyses:**

Not applicable. The paper is purely theoretical.

**Methods And Evaluation Criteria:**

This is a purely theoretical paper.

**Other Comments Or Suggestions:**

I caught some very minor grammar typos when reading the paper. So I would recommend the authors to make a revision in this aspect. (Eg: “we adopts” in the last paragraph of page 2.)

**Other Strengths And Weaknesses:**

The main strength of the paper is that it addresses an open problem proposed in a previous paper (Apers and de Wolf). Another interesting aspect is that the proposed solution comes by adapting and extending results obtained quite recently in neighboring areas (eg, the papers by Lee and Jambulapati , and the poper by Hamoudi 2022).

**Questions For Authors:**

No question

**Relation To Broader Scientific Literature:**

The paper refers to relevant literature in an appropriate way. Most of the main cited papers are recent, and are actually used as a basis for the development of quantum the new quantum algorithms. Notably, the paper from Apers and De Wolf, where the hypergraph sparsification problem is mentioned, the paper by Lee and Jambulapati which is used as a basis for the new quantum algorithm for hyperedge leverage score overestimate, and the paper by Hamoudi, which describes a technique to prepare multiple copies of a quantum state specified by an oracle.

**Theoretical Claims:**

I read carefully the first part of the paper, and skimmed through the appendix. I did not check carefully the correctness of the proofs in the appendix.

---

> ### Author Rebuttal · Authors · 2025-03-31
>
> Thank you for your helpful comments and suggestions. We will carefully
> revise our paper to correct all typos, especially we will change the
> word "adopts" in the last paragraph of page 2 to "adopt".

---

### Decision · Program_Chairs · 2025-05-01

**Decision:**

Accept (poster)

**Comment:**

The paper presents the first quantum algorithm for hypergraph sparsification, and demonstrates quantum speedups for computing hypergraph cut sparsifiers, and related combinatorial problems for hypergraphs. All the reviewers give positive evaluation on the paper, and  appreciate the theoretical contribution of the work. The reviewers mentioned a few minor questions, and the authors gave satisfactory answers to the questions. I would recommend acceptance of the paper.